# Current State of Natural Populations of *Paeonia anomala* (Paeoniaceae) in East Kazakhstan

Serik A. Kubentayev [1,2,*], Oxana N. Khapilina [3], Margarita Yu. Ishmuratova [4], Aisulu K. Sarkytbayeva [5,*], Ainur S. Turzhanova [3], Akzhunis A. Imanbayeva [6], Daniyar T. Alibekov [1,2] and Moldir Z. Zhumagul [1,2,5,*]

1  Astana Botanical Garden, Orunbur 16, Astana 010000, Kazakhstan
2  Department of Graduate School of Natural Sciences, Astana International University, Kabanbai Batyr 8, Astana 010000, Kazakhstan
3  National Center for Biotechnology, Qorghalzhyn Hwy 13, Astana 010000, Kazakhstan; oksfur@mail.ru (O.N.K.); turzhanova-ainur@mail.ru (A.S.T.)
4  Department of Botany, E.A. Buketov Karaganda University, St. Universitetskaya, 28, Karaganda 100028, Kazakhstan; margarita.ishmur@mail.ru
5  Faculty of Biology and Biotechnology, Department of Biodiversity and Bioresources, Al-Farabi Kazakh National University, Al-Farabi Avenue 71, Almaty 050040, Kazakhstan
6  Mangyshlak Experimental Botanical Garden, Microdistrict 10/2, Aktau 130000, Kazakhstan
*  Correspondence: kubserik@mail.ru (S.A.K.); sarkytbaeva.aisulu@gmail.com (A.K.S.); moldirzhumagul@gmail.com (M.Z.Z.); Tel.: +7-7478788917 (A.K.S.)

**Abstract:** *Paeonia anomala* L. is a valuable and sought-after medicinal plant for treating therapeutic pathologies. The natural habitat of *P. anomala* in the Republic of Kazakhstan is located in the mountainous areas of the East Kazakhstan region. *P. anomala* is listed in the Red Book of Kazakhstan as a rare species with limited distribution. In this regard, we studied a strategy for preserving the biological diversity of *P. anomala* wild population. In particular, the ecological, phytocenotic, and floristic characteristics of five *P. anomala* populations in East Kazakhstan were explored. The anatomical, morphological, and genetic variability of the species in various habitats was evaluated. Overall, the condition of the *P. anomala* population in the study region can be considered satisfactory. The floristic composition of *P. anomala* plant communities recorded 130 species belonging to 35 families and 101 genera. The northern slopes of mountains and shrub-grass communities with leached chernozem with high contents of $N-NO_3$ and $P_2O_5$ appeared to be optimal for *P. anomala* growth. Asteraceae (13%), Rosaceae (13%), Poaceae (10%), and Ranunculaceae (9%) are the major families of *P. anomala* flora and plant communities. The Eurasian (54%), Asian (24%), and Holarctic (15%) groups were recognized as dominant in the chronological spectrum. Amplification with iPBS primers resulted in the generation of 505 fragments, 257 of which were polymorphic. Our research results indicate that the genetic differentiation of the Kazakhstan populations is not quite as high and may indicate their long-term existence within one large population. A separate branch is formed by the P5 population, which is located separately from other populations, confirming its genetic isolation. The analysis of genetic diversity iPBS markers suggests the existence of a large, unified *P. anomala* population in Kazakhstan Altai.

**Keywords:** conservation of biodiversity; anatomical and morphological variability; genetic diversity; *Paeonia*; Kazakhstan; floristic composition of species communities

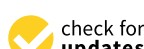



## 1. Introduction

The *Paeonia* L. genus is the only genus of the Paeoniaceae family, which includes 32 species of shrubs and perennial herbs distributed mainly in the northern hemisphere [1]. Kazakhstan flora [2,3] possesses two species of the genus *Paeonia*: *P. anomala* L. and *P. hybrida* Pall. [4]. Both represent a group of taxonomic complexes of herbaceous species of the *P. anomala* family, which also includes *Paeonia tenuifolia* L., *P. intermedia* C.A. Mey.,

*P. sinjiangensis* K.Y.Pan, *P. altaica* K.M.Dai and T.H.Ying, and *P. veitchii* Lynch, distributed in Siberia, Central Asia, and adjacent regions [5]. However, according to recent taxonomic revisions [1,6], only three species have been recognized as related to *P. anomala*: *P. anomala*, *P. intermedia*, and *P. tenuifolia*. The remaining species were reduced to synonyms. All species of *P. anomala* kinship complex are diploid [1,7,8].

Paeonia anomala is known worldwide as one of the most valuable plants because of its ornamental value and medicinal properties. It is a vital and essential drug in Eurasian countries, including Russia, Mongolia, and China. Monoterpene glycosides, flavonoids, tannins, stilbene, triterpenoids, and other compounds have been found in its roots [9–18]. These compounds have antioxidant, anti-inflammatory, and antipathogenic properties [13,19–24].

Peony roots contain a valuable natural compound, paeoniflorin, which has antitumor effects in various types of human cancers, including gliomas [25–29]. *P. anomala* is considered to be one of the most common medicinal herbs of Chinese traditional medicine for treating central nervous system diseases [26,30,31]. Peony roots are also used as a pain reliever for female disorders [19,32,33], stomach ailments, bleeding, exhaustion, and respiratory diseases, as well as for epilepsy and coughs [34].

*P. anomala* is a mesophyte with a wide distribution in the northern hemisphere, from the European part of Russia to Mongolia and China. The original ancestor of the peony is considered a small tree or shrub growing in humid and shady habitats [35]. The annual layers found in *P. anomala* rhizome [36] confirm a tree-type construction. The peony seed structure with an imperfect and underdeveloped embryo and multi-stored fruit, typical for relatively primitive genera and families, allows its attribution to relic components of the northern hemisphere flora [37]. In addition, the spiral arrangement and preservation of the peony ventral bundle confirm leaflet primitiveness [36].

The Katon-Karagai, Kurchum, Ulan, and Zyryanovskiy administrative districts in the East Kazakhstan region are the heart of the distribution of *P. anomala* in Kazakhstan [38–40]. The Zhambyl and Almaty regions are attributed to less extensive areas of distribution, where the most extreme points of the southwest border of the species' spread are marked. *P. anomala* is listed in the Red Book of Kazakhstan as a rare species with limited distribution [41]. In the study region, *P. anomala* is protected in the Katon-Karagai State National Nature Park and Markakol State Reserve. As a result of the intensive harvesting of *P. anomala* roots and flowers, wild populations are increasingly threatened and sharply reduced. Peony seed regeneration is usually low in natural habitats because seeds are damaged by weevil larvae and have delayed seed germination and seedling formation due to weak embryo development and low activity of basic enzymes [42].

Despite relatively numerous studies on the chemical composition and medicinal properties of *P. anomala*, the biological features and ecological–phytocoenotic confinement of the species remain poorly understood. Bioecological studies of *P. anomala* populations have been conducted mainly in CIS (Commonwealth of Independent States) countries [43–49] in local isolated territories. Studies of the genetic diversity of peony using PCR methods are not exhaustive enough, although scientific publications of their results indicate the use of different types of markers such as AFLP, SSR, ISSR, and RAPD and allow the elucidation of the structure and genetic relationships in populations of different peony species [24,50,51]. However, these studies have been conducted mainly on populations of other species located in China or South Korea [52–54], whereas studies of *P. anomala* natural populations were limited or did not fully cover the growing regions, and the genetic diversity of Kazakhstan populations had not previously been investigated. We suggest that DNA markers based on retrotransposons may be the most effective for solving this problem because this type of marker is widely used in various animal and plant species, including those that possess unknown genomes [55–58]. We believe that biodiversity analysis will improve our understanding of the genetic diversity, structure, and genetic relationships of *P. anomala* wild populations in Kazakhstan, which are increasingly threatened and dramatically reduced due to the intensive harvesting of roots and flowers.

Such investigations are conducted here for the first time, thereby supporting the relevance and novelty of this research. The aim of this study was to comprehensively analyze the current state of *P. anomala* populations in eastern Kazakhstan and assess their genetic biodiversity as well as floristic, anatomic–morphological, and ecological–phylogenetic characteristics of habitats. The research outcomes will contribute to the development of a meaningful strategy for the preservation of the natural diversity of *P. anomala* wild populations.

## 2. Materials and Methods

### 2.1. P. anomala Ecological and Phytocenotic Confinement, Morphology, and Floristic Studies of Populations

The research was conducted from 2018 to 2022 in three geographic sectors of East Kazakhstan: South, West, and Kalba Altai, according to physical–geographical zoning [59], where the Kazakhstan Altai represents a system of ranges in the southwest and southern part of this mountainous country, extending from south to north and from west to east for almost 400 km. The study region is the administrative region of East Kazakhstan. We used generally accepted geobotanical methods and applied an ecological–physiognomic approach to study the ecological–phylogenetic features of *P. anomala* populations. Ecological and physiognomic types unite plant communities with dominants belonging to the same eco-biomorphic index and ecological groups [60,61].

In each population, 15 plots were established for the study, and the area of each plot was: $10 \times 10$ m ($100$ m$^2$). A total of 75 plots were counted. GPS assisted in determining the boundaries of the communities. First, general information about the plot, such as description number, geographic location, date, coordinates, height, size, and photo number, was recorded in special forms of geobotanical description. After that, the following main sections were identified: the name of vegetation type based on dominant species; the floristic composition of the community with the assessment of species abundance on the Brown–Blanke abundance scale; and the morphological features of 20 replications.

The life cycles of peonies were studied according to the method of A.A. Uranov [62]. The ecological and biological features of the species were examined according to the methodological recommendations developed by V.N. Golubev and E.F. Molchanov [63]. The distribution of the floristic composition of *P. anomala* communities by ecological groups and species range was performed according to Kuminova's classification [64]. Plant names were given according to Plants of the World Online [65]. A total of 83 herbarium samples (NUR 005250—NUR 005333) were deposited in the herbarium of the Astana Botanical Garden (NUR).

The statistical analysis of morphometric parameters was performed using the program Statistica 10 (StatSoft STATISTICA 10, 2011).

Experimental and field studies of *P. anomala* populations, including the collection of plant material, were conducted in accordance with national and international directives and in strict compliance with the legislation of the Republic of Kazakhstan.

The identification of plant species was carried out by S.A. Kubentayev, and herbarium material was deposited in the herbarium fund of the Astana Botanical Garden (NUR).

The datasets analyzed in the present study are available from the author A.K. Sarkytbayeva upon request.

### 2.2. Variability in the Anatomical Structure of P. anomala Aboveground Organs

To study the anatomical structure of *P. anomala*, the aboveground parts of the peony were collected during the fruiting phase from each population. Leaves and stems were fixed in a mixture of glycerol, 96% alcohol, and distilled water in a 1:1:1 ratio (Strauss–Fleming mixture). Transverse sections of the leaf, petiole, and stem were prepared manually. For stem sections, microsections were obtained from the middle part of generative shoots; for leaf sections, microsections were obtained from the central part of the terminal leaf fragment; and for petiole sections, microsections were obtained from the middle part. Glycerol was used to enhance the clarity of the sections. The prepared samples were photographed using

an Altami microscope with a 3.1 Mpix digital camera at 16 × 4 and 16 × 10 magnifications. The processing of photographs and measurements of the micropreparations was performed using the Altami Studio program via Paint 10.0. To describe the anatomical structure, we used the principles outlined in the works of L.I. Lotova [66], G. Prenner, and P.J. Rudall [67]. A minimum of 10 micropreparations were created for each specimen.

### 2.3. Genetic Diversity of P. anomala Population

To analyze the genetic diversity of natural populations of *P. anomala*, we used the iPBS method. Plant specimens of the same age were randomly selected within the population, growing at a distance of more than 10 m from each other to ensure the reliability of the data obtained and ensuring minimal damage to living plants. For DNA extraction, each sample comprised an equivalent amount of fresh tissue originating from 1 of 10 randomly selected plants within a specific population. Genomic DNA was extracted from fresh plant leaves using modified acidic CTAB (cetrimoniumbromid) extraction buffer (2% CTAB, 2 M NaCl, 10 mM Na3EDTA, 100 mM HEPES, pH 5.3) with RNase A treatment [68].

The extracted DNA was dissolved in 100 μL of 1 × TE buffer (1 mM EDTA, 10 mM Tris-HCl, pH 8.0). DNA concentration was determined spectrophotometrically using a NanoDrop1000 spectrophotometer (Thermo Scientific).

The visualization of extracted DNA was performed on 1% agarose gel using the PharosFXPlus (Bio-RadLaboratories Inc., USA). DNA samples were prepared in two variants: uterine solution for long-term storage at −20 °C and working solutions for PCR at a concentration of 10 ng/μL.

The genetic diversity of the *P. anomala* populations was estimated using the inter-primer binding site (iPBS) retrotransposon markers (Table 1) [55].

**Table 1.** Characteristics of iPBS primers, used for *P. anomala* genetic diversity analysis.

| ID | Sequence | Tm (°C) | CG (%) | Optimal Annealing Ta (°C) |
|---|---|---|---|---|
| 2230 | tctaggcgtctgatacca | 54 | 50 | 52.9 |
| 2237 | cccctacctggcgtgcca | 65 | 72.2 | 55 |
| 2240 | aacctggctcagatgcca | 58.9 | 55.6 | 55 |
| 2253 | tcgaggctctagatacca | 53.4 | 50 | 51 |

The PCR reaction was conducted in 20 μL of reaction mixture, which included 3 μL of DNA (10 ng/μL), 4 μL Phire Reaction Buffer 5x with MgCl$_2$, 1 μL of primer (10 mM), 0.4 μL of dNTPs mixture (10 mM), and 0.2 μL of 1 U Phire Hot Start polymerase. The amplification protocol was as follows: pre-denaturation at 98 °C for 2 min, followed by 30 cycles of 98 °C for 30 s, 50–57 °C for 1 min, 72 °C for 1 min and additional elongation at 72 °C for 2 min. Amplification was performed in a SimpliAmp™ amplifier (Applied-Biosystems, Thermo Fisher Scientific, Singapore). Each DNA sample underwent at least two repetitions of PCR. The resulting PCR products were visualized on a 1.5% agarose gel with ethidium bromide. The sizes of the amplified DNA fragments were determined through comparison with a marker (Thermo Scientific GeneRuler DNA Ladder Mix 100–10,000 bp). Fragment lengths were determined using the Quantity One program in the PharosFXPlus gel documentation system (Bio-Rad Laboratories, Hercules, CA, USA).

For DNA fingerprinting, only clear and scorable bands were considered for analysis. Each band of distinct size corresponds to a unique genetic locus. In constructing a binary matrix, replicated fragments were recorded as "present" (1) or "absent" (0).

The detectable level of polymorphism was determined by calculating the percentage of polymorphic amplicons relative to the total number of amplicons for each primer. Key genetic diversity indicators, such as the number of alleles, the Shannon information index (I), and the index of genetic differentiation (PhiPT), were determined using GenAlex 6.5. The dendrogram was constructed using the UPGMA method [55].

## 3. Results

### 3.1. P. anomala Ecological and Phytocenotic Confinement and Morphology

In the study region, *P. anomala* exhibits pronounced ecological plasticity and grows in various communities in the altitudinal range of 700–1850 m above sea level. The species grows on shadow-side slopes, intermountain depressions, among bushes, and in sparse dark coniferous and pine forests. *P. anomala* habitats in the foothills are related to the forest and bushy slopes with northern exposure. In the highlands, the species reside in alpine and subalpine meadows and Siberian larch (*Larix sibirica* Ledeb.) and Siberian pine (*Pinus sibirica* Du Tour.) forests, and can also be found less often on uncultivated stony slopes. The shrub associations of *P. anomala* are formed in communities dominated by *Spiraea media* Schmidt, *Rosa spinosissima* L., *Rosa acicularis* Lindl., and *Lonicera caerulea* subsp. *altaica* (Pall.) Gladkova. In the areas of western and southern Altai, the species occurs in the dark coniferous forests of *Pinus sibirica* Du Tour and *Picea obovata* Ledeb. They also grow in soft deciduous forests dominated by *Betula pendula* Roth and *Populus tremula* L. Below are descriptions of the main populations of *P. anomala* in different regions of Kazakhstan Altai.

Southern Altai. Two populations of *P. anomala*, Belkaragai (P1) and Barlyk (P2) were identified and surveyed in the area of southern Altai (Figures 1 and 2). Both populations are located in the Katon-Karagai district of the East Kazakhstan region.

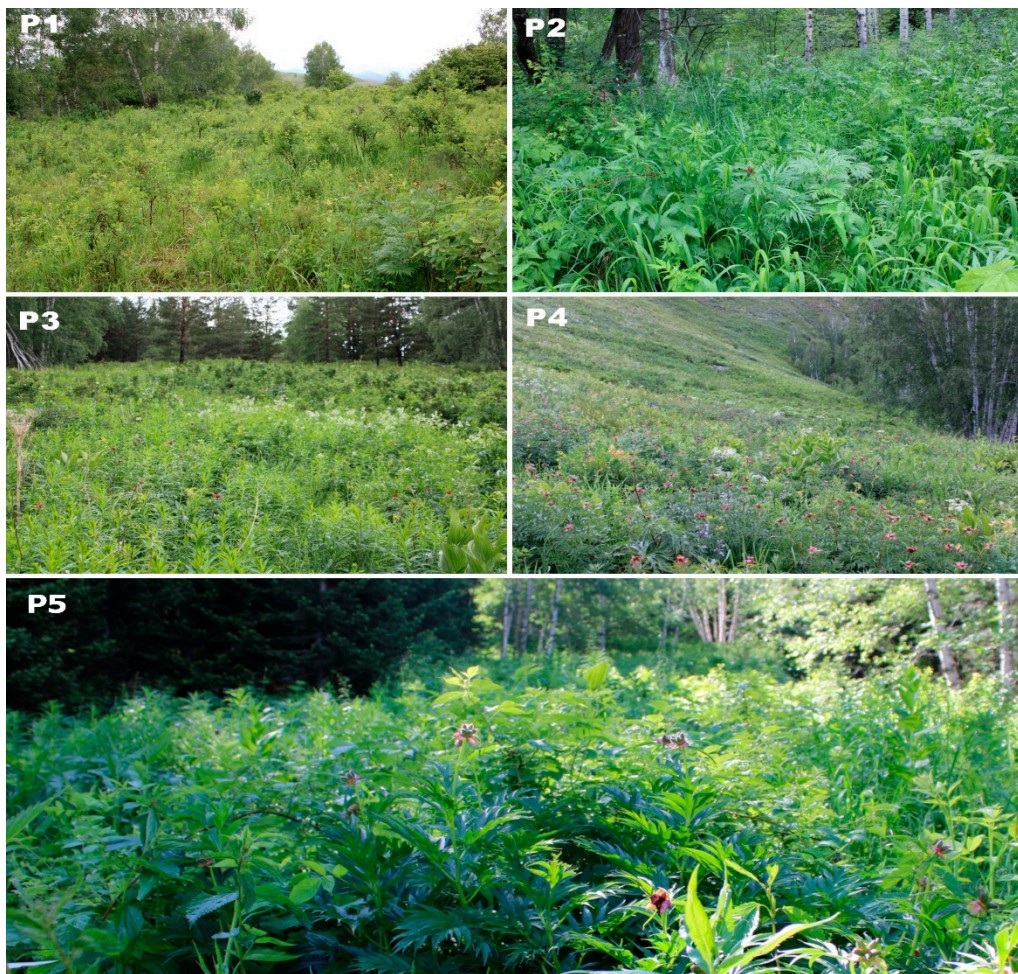

**Figure 1.** *P. anomala* populations in different ecological conditions (**P1–P5**) (photo by *S. Kubentayev*).

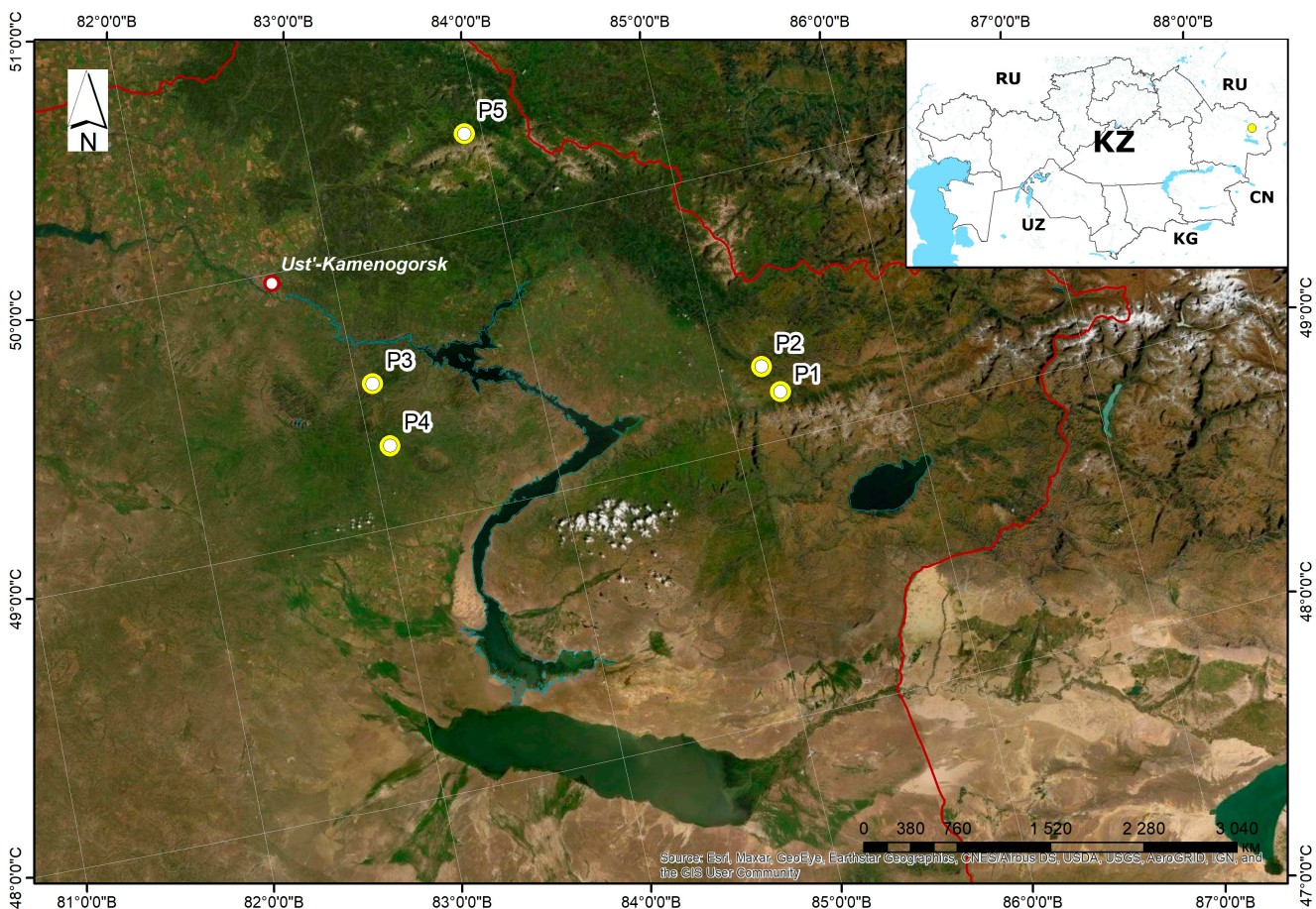

**Figure 2.** Geographic locations of *P. anomala* populations (P1–P5) (map was created using ArcGIS Desktop 10.8.2).

P1 is located on the Sarymsakty ridge 5 km south of the village Belkaragai. Coordinates: 49°10′25.9″ N 85°18′16.7″ E, H—987 m. The population resides on the northwestern slope of a foothill terrace with a slight slope (8°). The total projective cover is 80%, and *P. anomala* accounts for approximately 6%. The ground cover is formed by sediment 1.5 cm thick; pet droppings are often noted. The species grows as part of *Rosa spinosissima* L.—*P. anomala*—*Fragaria vesca* L.—*Poa pratensis* L. + *Filipendula vulgaris* Moench communities. The following species frequently occur in phytocenosis: *Artemisia sericea* Weber ex Stechm., *Campanula sibirica* L., *Dactylis glomerata* L., *Galium verum* L., and *Thalictrum simplex* L. The population is exposed to considerable anthropogenic loads (cattle grazing): individuals are trampled and die during the early stages of development. This factor affects the renewal of species in the population. In particular, generative and senile individuals dominate over young individuals of the virginal period, thereby defining the population as degenerating.

P2 is located 12 km southeast of Barlyk village, Bukhtarma valley, in the Katon-Kargai National Park. The coordinates are 49°16′41.8″ N 85°14′8.5″ E, H—898 m. The population resides on a relatively steep (20°), northeastern slope of the mountain. The total projective cover is 100%, and *P. anomala* occupies 5% of the cover. The ground cover is formed by fallen leaves and foliage, 2–2.5 cm thick. The species grows in a sparse birch–aspen (*Betula pendula* Roth, *Populus tremula* L.) forest; it is a part of *Bromus inermis* Leyss—*Poa pratensis* L.—*Dactylis glomerata* L.—*Filipendula ulmaria* (L.) Maxim.—*Geranium sylvaticum* L.—*Aconitum septentrionale* Koelle—*P. anomala* communities. Among the species associated with phytocenosis: *Carex acuta* L., *Rubus saxatilis* L., *R. idaeus* L., *Veratrum lobelianum* Bernh., *Spiraea chamaedryfolia* L., *Prunella vulgaris* L. Logging for the care of birch and aspen plantations negatively influences this *P. anomala* population: plants are severely damaged

by tree felling and transportation, which causes a subsequent decrease in the numbers of all age states of *P. anomala*, and young individuals cannot recover from the damage.

Kalba Altai. Two populations of *P. anomala*, Asubulakskaya (P3) and Taintinskaya (P4), located in the Ulan District of the East Kazakhstan Province, were examined in the Kalba Altai (Figures 1–3).

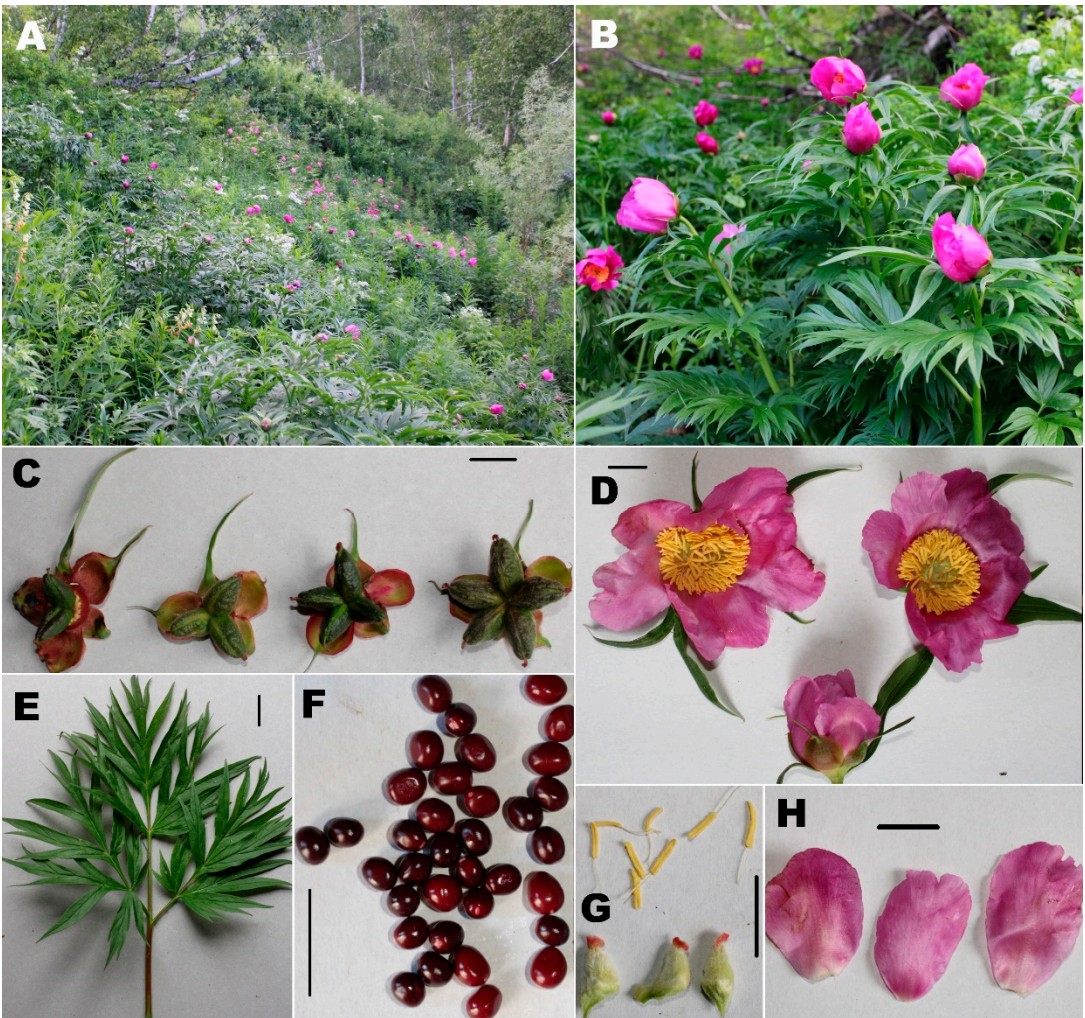

**Figure 3.** *P. anomala* external structure from the Kalba Altai ((**A**)—population; (**B**)—generative individual in the flowering phase; (**C**)—fruit (multiple leaflets) from 2 to 5 leaflets; (**D**)—flower; (**E**)—leaf; (**F**)—seeds; (**G**)—stamens and pistils; (**H**)—flower petals) (scale 1 cm).

P3 resides 10 km southeast of the village of Asubulak, on the western foothills of the Kalba Range coordinated within 49°31′53.22″ N, 83°5′14.98″ E, H—1136. The population occurs at the top of the mountain ridge; the microrelief is heterogeneous, formed of outcrops of large fragments of granite rocks and escarpments. The 12° slope faces northwest. The total projective cover is 100%, and *P. anomala* accounts for 15%. The ground cover comprises fallen leaves, needles, and the foliage of trees and is 3–3.5 cm thick. The species belong to *P. anomala* L.—*Rosa spinosissima* L.—*Epilobium angustifolium* L. communities along the edges of birch–pine (*Pinus sylvestris* L., *Betula pendula* Roth) forests. Secondary species in the community include *Aconitum leucostomum* Vorosch., *Anthriscus sylvestris* (L.) Hoffm., *Dracocephalum ruyschiana* L., *Filipendula vulgaris* Moench, *Fragaria vesca* L., *Serratula coronata* L., and *Polemonium caeruleum* L.

Peony plants in the surveyed population were not subjected to anthropogenic influence; therefore, no damage by diseases and pests was observed. Good seed and vegetative

reproduction of 4–6 young plants per 1 m$^2$ was noted. The population is complete, of the normal type, and able to hold the occupied area and to disperse to new territories.

P4 is located 15 km south of Tainty village, on the western foothills of the Kalba Range, coordinated in 49°17′49.9″ N, 83°6′42.52″ E, H—1152. The population is located on a steep northeastern slope (35°) of the mountain; the microrelief is leveled and slightly undulating. *P. anomala* accounts for about 35% of the total projective cover. The ground cover is formed by sediment 3–3.5 cm thick. The species comprise *P. anomala—Spiraea media* Schmidt communities. Associated species often include: *Agrostis gigantea* Roth, *Aconitum leucostomum* Worosch., *Dactylis glomerata* L., *Phleum pratense* L., *Saussurea latifolia* Ledeb., *Geranium pratense* L., *Lathyrus pratensis* L., and *P. anomala*, which form solid, clear thickets in places, and the individuals are powerfully developed multi-shrubs. In this population, anthropogenic influence was not observed, and populations of a normal type were observed in the presence of all age states. Habitat conditions in this population are optimal for species growth and development.

Western Altai. In the area of western Altai, near the village Poperechnoe, a population of *P. anomala*, denoted as P5, was examined (Figures 1 and 2). P5 is located on the Ivanovsky ridge, in the vicinity of the Gray Meadow tract. P5 coordinates within 50°21′08.8″ N, 83°53′47.8″ E, H-1195 m. It is on the northeastern slope of the mountain (slope 18°), at the lower limit of the forest. *P. anomala* projective cover accounts for 20%. The ground cover is formed by sediment of 1.5–2 cm thickness. The species is related to *Saussurea latifolia* Ledeb. + *P. anomala* communities along the edges of a spruce (*Picea obovata* Ledeb.) forest with *Betula pendula* Roth and *Abies sibirica* Ledeb. This community includes *Agrostis gigantea* Roth, *Veratrum lobelianum* Bernh., *Epilobium angustifolium* L., *Phlomoides alpina* (Pall.) Adylov, Kamelin Makhm., *Delphinium elatum* L., and *Filipendula ulmaria* (L.) Maxim. The *P. anomala* population ascends to the subalpine belt at a height of 1800 m on the Ivanovsky ridge and is part of the *Trollius altaicus* C.A. Mey. + *Aquilegia glandulosa* Fisch. ex. Link + *P. anomala* communities. The population is considered normal with a predominance of generative individuals, and it is not exposed to anthropogenic influence. The species in the given population reproduce mainly through vegetative means. Seed reproduction is poorly represented, and dense grasses displace young individuals during the early stages of development.

The populations of *P. anomala* in the area of southern Altai (P1, 2) are depressed due to anthropogenic impacts in the form of cattle grazing (P1) and logging (P2). These factors negatively affect the renewal of the species and are expressed in a small number of young individuals. The state of the species populations in the Kalba Altai (P3, 4) is close to optimum; these populations can hold the area they occupy and disperse to new territories. In the area of western Altai (P5), the state of *P. anomala* populations is normal, with a predominance of generative individuals.

Table 2 shows the morphological and quantitative indicators of the examined populations of *P. anomala*. According to the data obtained, the highest height and largest diameter of an adult bush was noted in P5 (107.6 ± 2.94 cm and 99.2 ± 5.96 cm), and the lowest height of an individual was identified in P1 (78.6 ± 2.11 cm), while the smallest bush diameter was found in P2 (72.6 ± 4.10 cm). The number of generative shoots per bush ranged from 2.73 ± 0.35 pcs (P2) to 13.53 ± 1.61 pcs (P4). In all surveyed populations, the number of vegetative shoots per adult was not significant (1–3 pcs). In *P. anomala* populations growing in meadow communities and among shrubs, individuals had a multi-shoot structure (6–18 pcs), and under gentle stands, in thin forests, the bushes were loose and had few shoots (3–5 pcs). In the surveyed populations, (1) the number of fruit leaflets was usually 5 or 3, and much less often, it was 4 or 2; (2) the number of individuals differed: the highest number of adults per 100 m$^2$ was seen in P4 (11.2 ± 1.35 units), whereas the lowest was noted in P1 (4.93 ± 0.48 units). The greatest biomass of the aboveground part of a single individual was found in P4 (1.81 ± 0.05 kg), due to the greatest number of generative shoots. The lowest biomass of an aboveground part of a single plant was found in P2 (0.29 ± 0.03 kg). A similar situation was observed for the root biomass of a single adult,

where the greatest mass was noted in P4 (2.69 ± 0.25 kg) and the lowest was found in P2 (0.98 ± 0.09 kg) (Figure 3).

**Table 2.** Quantitative and morphological indicators of *P. anomala* specimens from the study plots P1–P5.

| No. | Quantitative and Morphological Characteristics | | *P. anomala* Study Populations | | | | |
|---|---|---|---|---|---|---|---|
| | | | P1 | P2 | P3 | P4 | P5 |
| 1 | Height of generative individuals at the time of fruiting (cm) | | 78.6 ± 2.11 | 81.1 ± 3.75 | 86.3 ± 2.45 | 96.8 ± 2.87 | 107.6 ± 2.94 |
| 2 | Diameter of an adult bush (cm) | | 86.1 ± 4.55 | 72.6 ± 4.10 | 73.6 ± 5.71 | 87.6 ± 3.97 | 99.2 ± 5.96 |
| 3 | Number of shoots per bush (pcs) | gen. | 6.45 ± 0.62 | 2.73 ± 0.35 | 5.26 ± 0.65 | 13.53 ± 1.61 | 7.66 ± 0.76 |
| | | veg. | 1.93 ± 0.34 | 1.00 ± 0.25 | 0.93 ± 0.22 | 0.66 ± 0.25 | 1.86 ± 0.32 |
| 4 | Number of leaves on one generative shoot (pcs) | | 9.62 ± 0.24 | 9.06 ± 0.44 | 8.50 ± 0.32 | 9.13 ± 0.36 | 9.83 ± 0.28 |
| 5 | Stem diameter (mm) | | 9.33 ± 0.45 | 8.33 ± 0.31 | 10.1 ± 0.52 | 10.3 ± 0.46 | 11.1 ± 0.39 |
| 6 | Leaf blade size from the middle part of the shoot (cm) | length | 14.7 ± 0.53 | 20.4 ± 0.57 | 19.5 ± 0.68 | 16.8 ± 0.48 | 24.4 ± 0.63 |
| | | width | 20.5 ± 0.55 | 27.0 ± 1.02 | 28.3 ± 0.68 | 21.9 ± 0.73 | 31.9 ± 0.58 |
| 7 | Leaf petiole size (cm) | length | 8.00 ± 0.36 | 9.15 ± 0.28 | 9.44 ± 0.47 | 10.2 ± 0.31 | 10.9 ± 0.29 |
| | | thickness | 0.22 ± 0.01 | 0.32 ± 0.02 | 0.42 ± 0.02 | 0.46 ± 0.02 | 0.48 ± 0.03 |
| 8 | Fruit size (multi-leaf) (cm) | diameter | 5.45 ± 0.30 | 6.10 ± 0.30 | 5.68 ± 0.18 | 5.43 ± 0.20 | 5.23 ± 0.18 |
| | | thickness | 1.11 ± 0.04 | 1.41 ± 0.06 | 1.05 ± 0.04 | 1.17 ± 0.06 | 1.12 ± 0.03 |
| 9 | Number of leaflets of the fruit (multiple leaflets) (pcs) | | 4.26 ± 0.22 | 4.22 ± 0.17 | 3.86 ± 0.27 | 4.13 ± 0.23 | 3.80 ± 0.26 |
| 10 | Number of generative individuals per 100 m$^2$ (pcs) | | 4.93 ± 0.48 | 5.13 ± 0.59 | 6.6 ± 0.62 | 11.2 ± 1.35 | 7.21 ± 0.63 |
| 11 | Raw biomass of the aboveground part of one generative individual (kg) | | 0.59 ± 0.04 | 0.29 ± 0.03 | 0.54 ± 0.04 | 1.81 ± 0.05 | 1.22 ± 0.52 |
| 12 | Weight of raw root of one generative individual (kg) | | 1.10 ± 0.24 | 0.98 ± 0.09 | 1.32 ± 0.11 | 2.69 ± 0.25 | 1.95 ± 0.18 |

The results of the chemical analysis from the soils of *P. anomala* study populations (Table 3) showed a high content of nitrate nitrogen, more than 30 mg/kg, in all samples. The highest N-NO$_3$ values (52.9–58.9 mg/kg) were observed in P2, P4, and P5. The content of mobile phosphorus (P$_2$O$_5$) was low (7.7–12.6 mg/kg) in P1, P3, and P5, whereas it was average in P2 (23.9 mg/kg) and P4 (18.7 mg/kg). Exchangeable potassium (K$_2$O) availability was elevated (395 and 530 mg/kg) in P1 and P5, whereas it was high in P2 (749 mg/kg), P3 (711 mg/kg), and P4 (676 mg/kg).

**Table 3.** Content of nutrients in the soil.

| No. | Population | Soil Layer | N-NO$_3$, mg/kg | P$_2$O$_5$, mg/kg | K$_2$O, mg/kg | Humus, % | pH |
|---|---|---|---|---|---|---|---|
| 1 | P1 | 0–30 | 35.5 | 9.7 | 395 | 9.94 | 7.14 |
| 2 | P2 | 0–30 | 55.0 | 23.9 | 749 | 10.39 | 7.88 |
| 3 | P3 | 0–30 | 35.9 | 7.7 | 711 | 7.32 | 6.37 |
| 4 | P4 | 0–30 | 52.9 | 18.7 | 676 | 5.28 | 5.42 |
| 5 | P5 | 0–30 | 58.9 | 12.6 | 530 | 5.30 | 5.87 |

Soil pH had different values: P4 and P5 were characterized by slightly acidic soil (5.42 and 5.87), P3 by medium acidic soil (6.37), and P1 and P2 by neutral soil (7.14–7.88). The content of organic matter (humus) varied from medium (P4, P5) to elevated (P3) and high (P1, P2). Thus, the soils in P4 and P5 comprise leached chernozem, whereas those in P1, P2, and P3 comprise meadow chernozem.

The data showed that *P. anomala* biomorphology (height, bush width, number of shoots per individual) is significantly influenced by the content of N-NO$_3$ and P$_2$O$_5$ in the soil. In P4 and P5 populations with high N-NO$_3$ and P$_2$O$_5$ contents, the highest height of individuals (96.8 $\pm$ 2.87 and 107.6 $\pm$ 2.94 cm) was found, and a large bush diameter (87.6 $\pm$ 3.97 and 99.2 $\pm$ 5.96 cm) and number of shoots per bush (13.53 $\pm$ 1.61, 7.66 $\pm$ 0.76 pcs) were observed. Simultaneously, the humus content and pH in these populations were lower than those in other populations. Thus, the optimal soils for the growth of *P. anomala* are leached chernozem with a high content of N-NO$_3$ and P$_2$O$_5$.

### 3.2. Floristic Composition of the P. anomala Communities

As a result of literature data processing, field studies, and herbarium materials in the floristic composition of plant communities, 130 species of *P. anomala* were recorded, belonging to 35 families and 101 genera (Appendix A).

The fractional classification of ecological groups allowed the analysis of the composition of the flora of *P. anomala* plant communities and revealed the dominance of mesophytes (60%), mesoxerophytes (12%), and hygromesophytes (10%) for 108 groups of species (83%) out of the total species composition.

The systematic analysis of the flora composition of *P. anomala* plant communities showed that, regarding the number of species, the major families were Asteraceae Bercht. J. Presl. (17 species, 13%), Rosaceae Juss. (17 species, 13%), Poaceae Barnhart (15 species, 10%), Ranunculaceae Juss. (12 species, 9%), Fabaceae Lindl. (7 species, 5%), Apiaceae Lindl. (6 species, 4%), and Lamiaceae Martinov (6 species, 4%). They account for 80 (61%) species of the total composition of the community flora.

A chorological analysis of the composition of *P. anomala* plant communities reveald that the Eurasian group (54%), the Asian group (24%), and the Holarctic group (15%) are represented the most, whereas the proportion of cosmopolitan group was relatively low (5%).

In all surveyed populations, mesophytic plants were the most common: *Sanguisorba officinalis, Trifolium pratense, Veratrum lobelianum, Vicia sepium, V. cracca, Poa pratensis, Polemonium caeruleum, Dactylis glomerata, Anthriscus sylvestris, Thalictrum flavum, Galium boreale, Aconitum leucostomum, Epilobium angustifolium, Geranium pratense* and *Filipendula ulmaria*.

### 3.3. Variability in the Anatomical Structure of P. anomala Aboveground Organs

Regarding leaf anatomical structure, *P. anomala* leaves are large, glabrous, long-petioled, and bipartite. The terminal leaflets are deeply pinnatipartite, with lanceolate elongate lobes; the margins are entire. On cross-section, the leaf is flat and dorsoventral (Figure 4). Leaf veins protrude considerably from the underside of the leaf. A leaf is surrounded by cells of the upper and lower epidermis on both sides; it is shaped with thickened outer walls. The epidermal surface is smooth; single simple trichomes on the upper side, mainly in the area of the middle vein, are observed in some plant specimens. Thus, trichomes were noted in plants from the P1, P4, and P5 populations, whereas no trichomes were noted in P2 and P3 leaf samples. The columnar mesophyll consists of a single layer of cells, whereas the spongy mesophyll is multilayered. The conductive bundle is collateral, of the closed type. The xylem is oriented toward the upper epidermis and the phloem is oriented toward the lower side. The Xylem zone is surrounded by mechanical tissue sclerenchyma.

Regarding stem anatomical structure, the *P. anomala* stem has a typical anatomical structure of dicotyledonous plants. In the cross-section, the stem is rounded, smooth, and without pubescence (Figure 5).

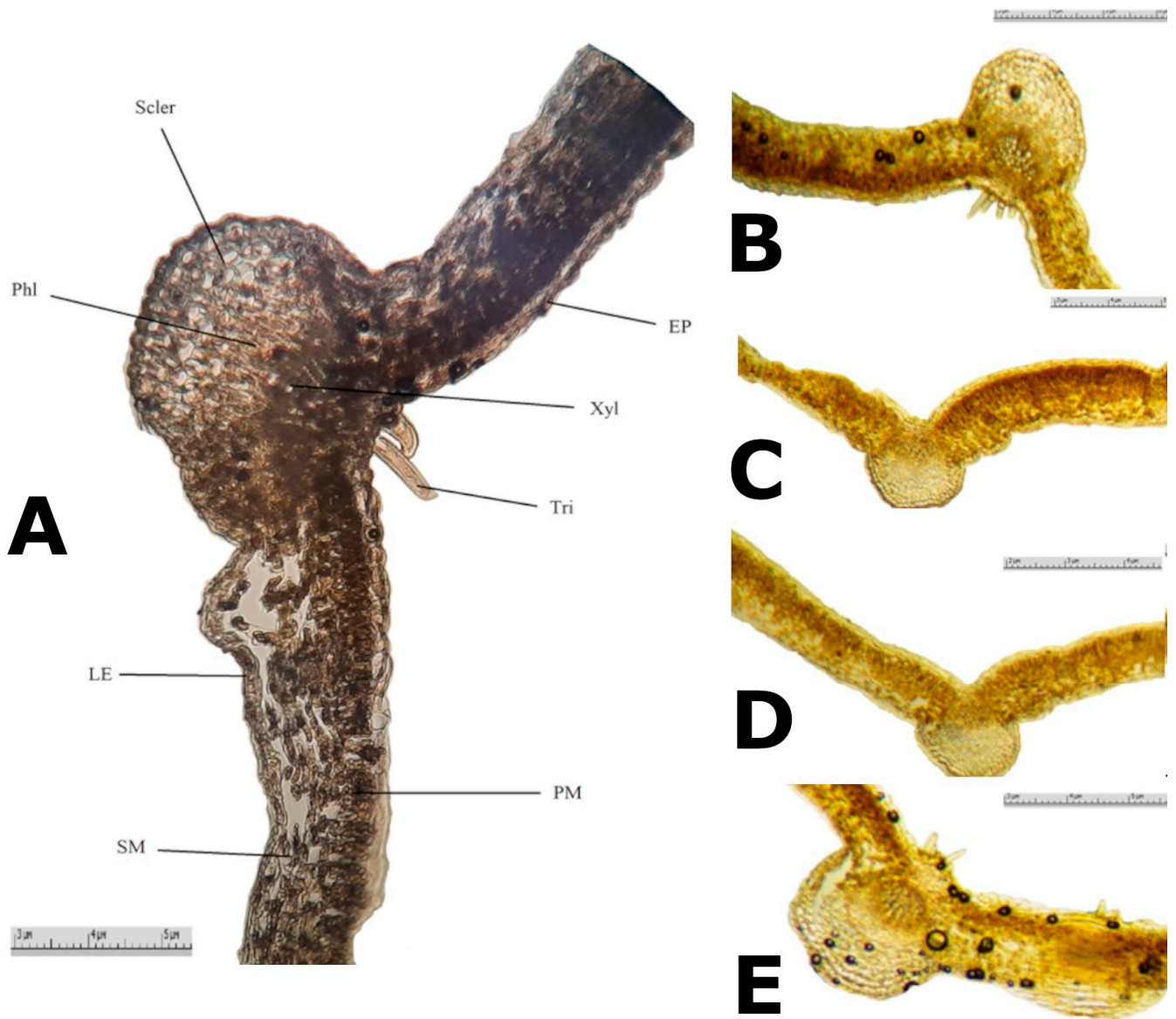

**Figure 4.** Cross-section of *P. anomala* leaf. Fragment in the area of the middle vein ((**A**)—P5 population. Mag. 16 × 20); photomicrographs leaf cross-sections of different *P. anomala* populations. Mag. 16 × 20. ((**B**)—P4 population; (**C**)—P3 population; (**D**)—P2 population; (**E**)—P1 population). UE—upper epidermis; LE—lower epidermis; Tri—trichomes; Xyl—xylem; Phl—phloem; CM—columnar mesophyll; SM—spongy mesophyll; Collen—collenchyma.

A single-layer epidermis surrounds the stem perimeter. Its cells are rounded, with considerably thickened outer cells and cuticles, the layer of which is thicker in the joints of the cells. Beneath the epidermis, there is a single-layer collenchyma, under which a multilayered chlorenchyma lies (2 to 6 layers). The conductive zone is of the bundle or transitional type. The fascicles are large, oval or broadly ovate, collateral, and of closed type (no cambium). The bundle is reinforced on both sides by sections of sclerenchyma. The inner part is filled with loose and thin-walled cells of the medullary parenchyma, which can break down to form air-carrying cavities.

Leaf petiole anatomical structure. *P. anomala* is characterized by grooved radial-type petioles with well-developed veins in the ventral part (Figure 6).

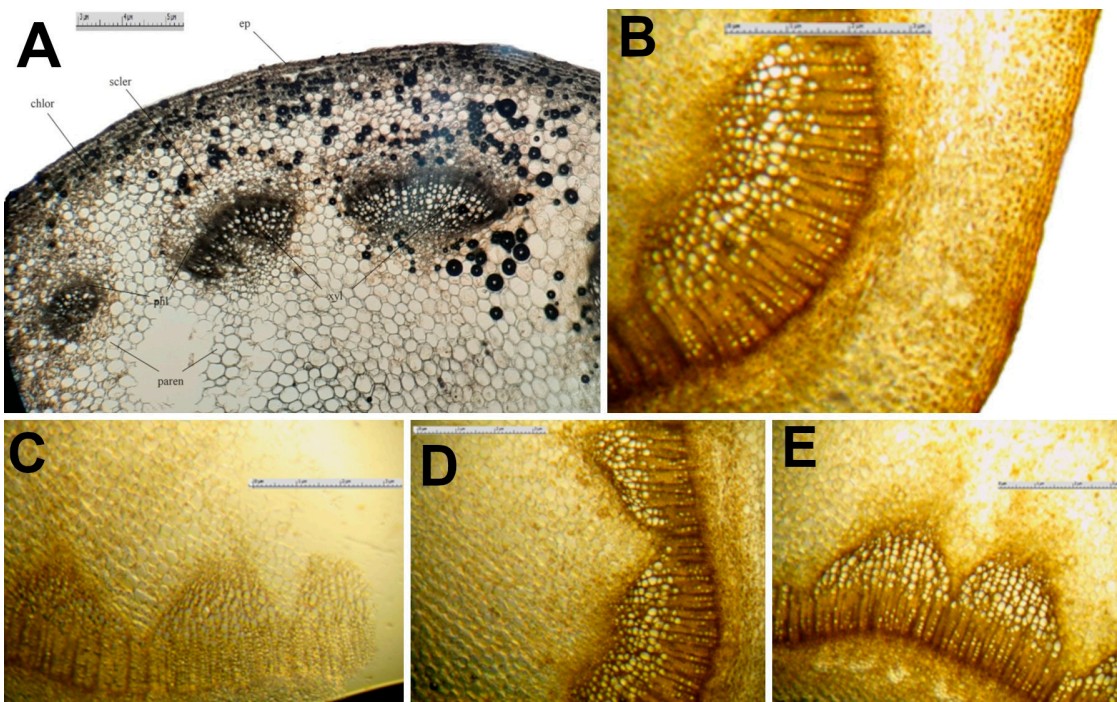

**Figure 5.** *P. anomala* stem cross-section ((**A**)—fragment. P5 population. Mag. 16 × 4); photomicrographs of fragments of transverse sections of *P. anomala* stem from different populations. Mag. 16 × 4. ((**B**)—P1 population; (**C**)—P2 population; (**D**)—P3 population; (**E**)—P4 population). Ep—epidermis; chlor—chlorenchyma; phl—phloem; xyl—xylem; paren—cord parenchyma; scler—sclerenchyma.

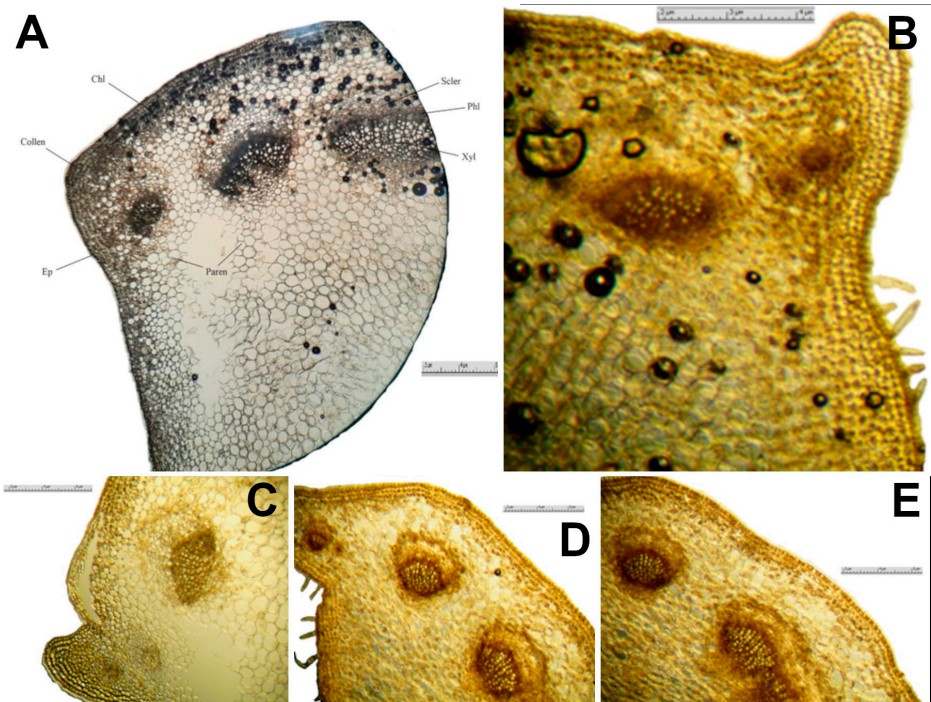

**Figure 6.** *P. anomala* leaf petiole cross-section. Lateral fragment. ((**A**)—P5 population. Mag. 16 × 20); photomicrographs of fragments of cross-sections of leaf petioles of *P. anomala* from different populations. Mag. 16 × 4. ((**B**)—P4 population; (**C**)—P3 population; (**D**)—P2 population; (**E**)—P1 population). Ep—epidermis; Collen—collenchyma; Chl—chlorenchyma; Scler—sclerenchyma; Phl—phloem; Xyl—xylem; Paren—core parenchyma.

The abaxial (dorsal) and adaxial (ventral) sides of the petioles are well expressed. The shape of the cross-section is curved with thinned lateral ears. The covering tissue is composed of small, distinctly shaped epidermal cells. The outer wall of the epidermis is thicker than the inner wall and is covered by a cuticle layer. The adaxial side has rare simple multicellular trichomes. The epidermal cells have an assimilatory tissue, the chlorenchyma, composed of 2–3 rows of cells. Chlorenchyma cells are thin-walled, rounded, or elliptical, and may vary in size. Collateral conductive bundles are the closed type with varying shapes and sizes. Lateral conductive bundles appear well developed and are located in the petiole abaxial part; they are usually broadly ovate. The abdomen bundles are small in size and oval or elliptical. Xylem in bundles faces toward the center, phloem toward the periphery. "Caps" of sclerenchyma mechanical tissue are marked above the fascicles. The interfascicular parenchyma is not woody. Parenchyma cells form the petiole central part, which may collapse into old leaves, forming cavities.

The comparison of leaf anatomical parameters, leaf petiole, and stem collected in different populations reveals differences in the thickness of tissue sections and individual cells (Tables 4–6).

**Table 4.** Quantitative indices of individual cells and tissues of *P. anomala* leaf from different populations (in μm).

| Indicators | *P. anomala* Populations | | | | |
|---|---|---|---|---|---|
| | **P1** | **P2** | **P3** | **P4** | **P5** |
| The thickness of leaf at midrib | $5.88 \pm 0.05$ [ns] | $5.89 \pm 0.006$ * | $6.14 \pm 0.03$ * | $6.11 \pm 0.05$ * | $5.91 \pm 0.02$ * |
| Leaf width on the side | $2.98 \pm 0.004$ [ns] | $3.00 \pm 0.04$ [ns] | $3.15 \pm 0.01$ [ns] | $3.12 \pm 0.04$ [ns] | $3.09 \pm 0.03$ [ns] |
| Diameter of the conductive bundle | $2.31 \pm 0.02$ * | $2.41 \pm 0.004$ * | $2.58 \pm 0.06$ [ns] | $2.54 \pm 0.02$ * | $2.46 \pm 0.01$ * |
| The thickness of the lower epidermis | $0.32 \pm 0.02$ [ns] | $0.34 \pm 0.008$ [ns] | $0.40 \pm 0.030$ [ns] | $0.37 \pm 0.005$ [ns] | $0.36 \pm 0.005$ [ns] |
| The thickness of the upper epidermis | $0,40 \pm 0.005$ * | $0.41 \pm 0.005$ * | $0.51 \pm 0.005$ * | $0.46 \pm 0.006$ [ns] | $0.45 \pm 0.004$ * |
| Trichome length | $1.31 \pm 0.004$ * | - | - | $1.42 \pm 0.008$ * | $1.36 \pm 0.005$ * |
| Columnar mesophyll thickness | $0.97 \pm 0.006$ * | $0.94 \pm 0.004$ * | $1.14 \pm 0.004$ * | $1.12 \pm 0.006$ * | $1.09 \pm 0.004$ * |
| The thickness of spongy mesophyll | $1.42 \pm 0.005$ * | $1.40 \pm 0.006$ * | $1.61 \pm 0.005$ * | $1.58 \pm 0.005$ * | $1.55 \pm 0.003$ * |
| Phloem thickness in the conductive bundle | $0.51 \pm 0.003$ * | $0.48 \pm 0.02$ * | $0.64 \pm 0.004$ * | $0.59 \pm 0.003$ * | $0.54 \pm 0.002$ * |
| Xylem thickness in the conductive bundle | $1.60 \pm 0.04$ [ns] | $1.57 \pm 0.006$ * | $1.70 \pm 0.008$ * | $1.68 \pm 0.008$ [ns] | $1.64 \pm 0.02$ * |

*—shows a reliable difference in a number of measurements ($p < 0.05$); ns—no reliable difference in the results of measurements ($p > 0.05$).

**Table 5.** Measurements of stem individual cells and tissues of *P. anomala* from different populations (in μm).

| Indicators | *Paeonia anomala* Populations | | | | |
|---|---|---|---|---|---|
| | **P1** | **P2** | **P3** | **P4** | **P5** |
| Epidermis thickness | $1.65 \pm 0.0078$ * | $1.70 \pm 0.06$ [ns] | $1.86 \pm 0.04$ [ns] | $1.82 \pm 0.01$ [ns] | $1.78 \pm 0.03$ [ns] |
| Chlorenchyma thickness | $2.80 \pm 0.006$ * | $2.85 \pm 0.007$ * | $3.12 \pm 0.006$ * | $2.95 \pm 0.051$ [ns] | $2.89 \pm 0.007$ [ns] |
| Conductive beam length | $10.2 \pm 0.09$ * | $10.5 \pm 0.15$ [ns] | $12.8 \pm 0.22$ * | $11.3 \pm 0.09$ * | $10.6 \pm 0.16$ * |
| Conductive beam width | $11.98 \pm 0.18$ * | $12.35 \pm 0.20$ [ns] | $13.61 \pm 0.16$ * | $12.08 \pm 0.23$ [ns] | $12.44 \pm 0.11$ [ns] |

*—shows a reliable difference in a number of measurements ($p < 0.05$); ns—no reliable difference in the results of measurements ($p > 0.05$).

**Table 6.** Measures of *P. anomala* leaf petiole individual cells and tissues from different populations (in μm).

| Indicators | *P. anomala* Populations | | | | |
| --- | --- | --- | --- | --- | --- |
| | P1 | P2 | P3 | P4 | P5 |
| Leaf petiole thickness | $6.20 \pm 0.09$ [ns] | $6.09 \pm 0.009$ [ns] | $6.92 \pm 0.11$ [*] | $6.28 \pm 0.08$ [ns] | $6.15 \pm 0.04$ [ns] |
| Epidermis thickness | $0.41 \pm 0.011$ [ns] | $0.40 \pm 0.008$ [ns] | $0.39 \pm 0.07$ [ns] | $0.42 \pm 0.11$ [ns] | $0.35 \pm 0.06$ [ns] |
| Chlorenchyma thickness | $0.83 \pm 0.016$ [ns] | $0.72 \pm 0.014$ [*] | $0.80 \pm 0.024$ [ns] | $0.79 \pm 0.013$ [ns] | $0.82 \pm 0.017$ [ns] |
| Side conductive bundle length | $2.20 \pm 0.024$ [*] | $1.93 \pm 0.010$ [*] | $1.98 \pm 0.05$ [*] | $2.15 \pm 0.014$ [*] | $2.11 \pm 0.008$ [*] |
| The thickness of the lateral conductive bundle | $1.35 \pm 0.030$ [*] | $1.06 \pm 0.012$ [ns] | $1.20 \pm 0.05$ [ns] | $1.32 \pm 0.14$ [ns] | $1.11 \pm 0.025$ [ns] |
| Trichome length | $1.36 \pm 0.023$ [ns] | $1.14 \pm 0.009$ [*] | $1.33 \pm 0.016$ [ns] | $1.22 \pm 0.015$ [*] | $1.32 \pm 0.009$ [*] |

*—shows a reliable difference in a number of measurements ($p < 0.05$); ns—no reliable difference in the results of measurements ($p > 0.05$).

### 3.4. Genetic Diversity of P. anomala Population

The amplification of *P. anomala* DNA resulted in clearly distinguishable amplicons, which varied in number, depending on the primer used (Figure 7).

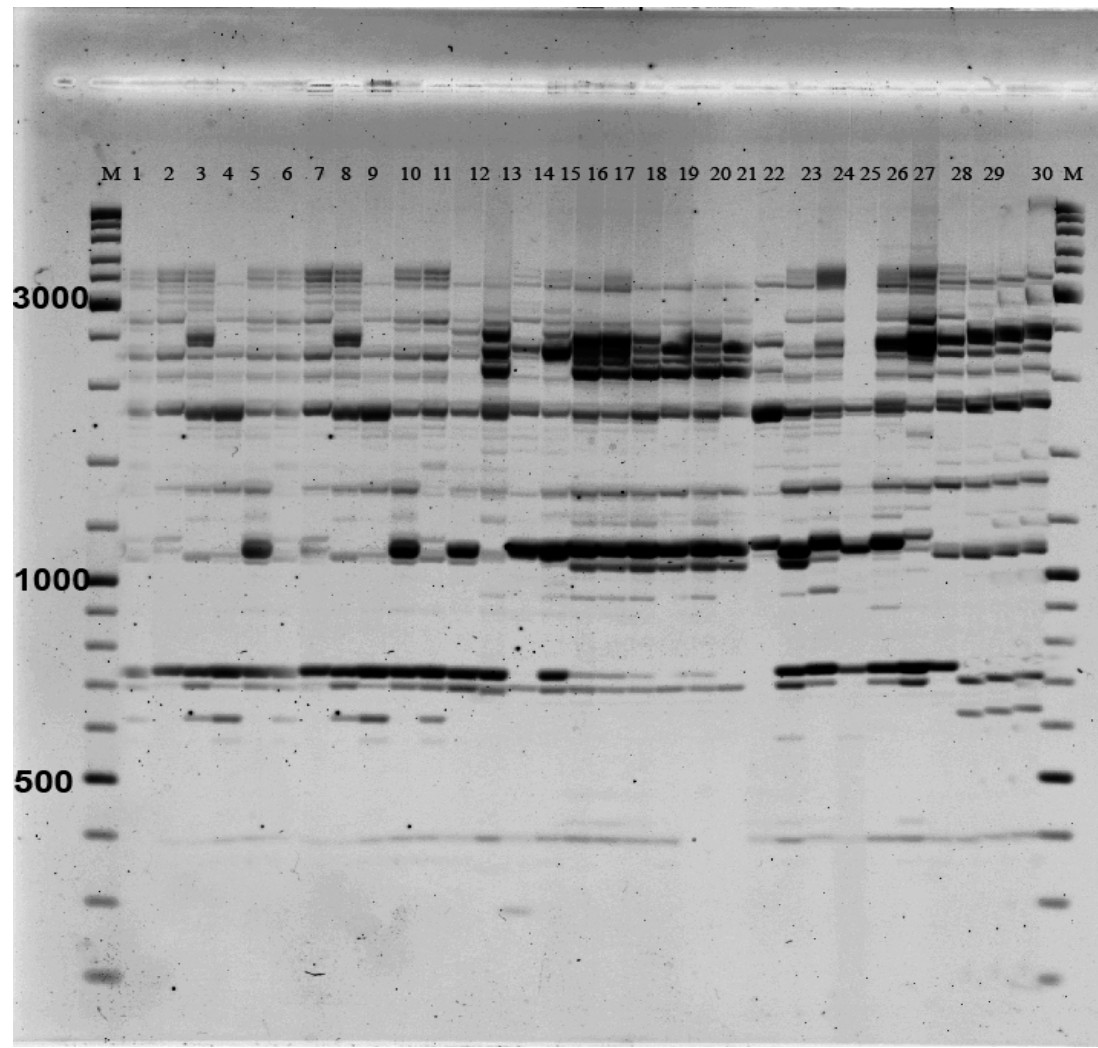

**Figure 7.** Electropherogram of amplification results of individual DNA samples from *P. anomala* populations with primer 2256. Samples from populations P1 (1–6), P2 (7–12), P3 (13–18), P4 (19–24), P5 (25–30); M—Thermo Scientific GeneRuler DNA Ladder Mix (100–10,000 bp).

To study the genetic variability of *P. anomala* wild populations, only clear scorable bands were considered since each band of unique size corresponds to a unique locus. The presence or absence of a band in the amplification profiles of a particular genotype was used to construct a binary matrix. Reproducible fragments were scored as present (1) or absent (0).

On average, each primer generated 126 fragments, 51% of which appeared polymorphic. Fragment sizes ranged from 500 to 3500 bp. The use of primers influenced the variety of polymorphism levels, from 46% (primer 2230) to 56% (primer 2253) (Table 7).

**Table 7.** Amplification result of iPBS primers, used for *P. anomala* genetic diversity analysis.

| ID | Sequence | Total Number of Loci | Number of Polymorphic Loci (%) |
|---|---|---|---|
| 2230 | tctaggcgtctgatacca | 132 | 46 |
| 2237 | cccctacctggcgtgcca | 138 | 51 |
| 2240 | aacctggctcagatgcca | 103 | 49 |
| 2253 | tcgaggctctagatacca | 132 | 56 |
| Average | | 126 | 51 |

Each peony sample was distinguished by a unique amplification profile, which allowed the determination of indicators of genetic diversity, reflecting the level of variability in the study populations based on DNA fingerprint data (Table 8).

**Table 8.** Genetic diversity of *P. anomala* populations through amplification with iPBS primers.

| Population | Na | Ne | I | He | uHe |
|---|---|---|---|---|---|
| P1 | 1.056 | 1.258 | 0.238 | 0.156 | 0.170 |
| P2 | 0.796 | 1.171 | 0.156 | 0.102 | 0.111 |
| P3 | 1.167 | 1.196 | 0.211 | 0.131 | 0.143 |
| P4 | 1.148 | 1.296 | 0.271 | 0.179 | 0.195 |
| P5 | 1.074 | 1.286 | 0.255 | 0.169 | 0.184 |
| Average | 1.048 | 1.241 | 0.226 | 0.147 | 0.161 |

The evaluation of genetic diversity of *P. anomala* populations revealed the maximum number of amplified Na fragments in the P3 population (1.167), whereas the minimum value of this index (0.796) was in the P2 population. The ranking of populations according to the Shannon diversity index in descending order is as follows: P4 (0.271) > P5 (0.255) > P1 (0.238) > P3 (0.211) > P2 (0.156).

The results of molecular variance (AMOVA) (Table 9) demonstrated that the majority (67%) of the molecular variability in *P. anomala* was due to intrapopulation variability (64%), whereas the level of interpopulation variability was only 34%.

**Table 9.** Results of AMOVA analysis of *P. anomala* populations via iPBS amplification.

| Variability | df | SS | MS | Est. Var. | % | $\phi$PT |
|---|---|---|---|---|---|---|
| Between populations | 4 | 100,533 | 25,133 | 3243 | 36% | 0.364 |
| Within populations | 25 | 141,833 | 5673 | 5673 | 64% | - |
| General | 29 | 242,367 | 30,806 | 8917 | 100% | - |

The results of the permutation test were used to estimate the probability of error and indicated the significance and reliability of differences between populations ($\phi$PT = 0.364, $p < 0.001$).

The data obtained by amplification with iPBS primers from different *P. anomala* populations were used for cluster analysis and the construction of the UPMGA dendrogram (Figure 8).

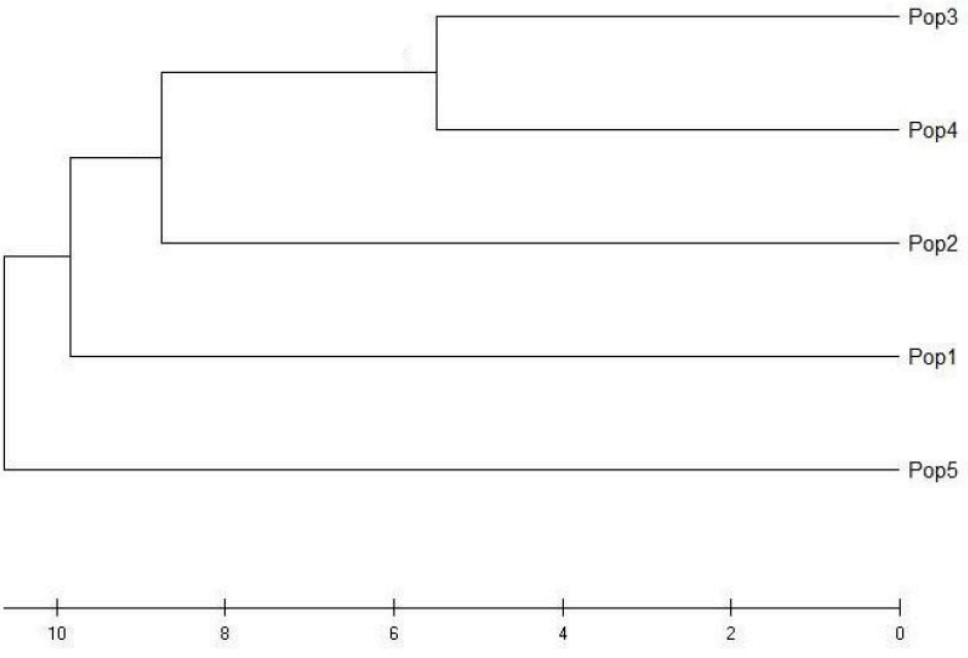

**Figure 8.** Dendrogram of genetic distances of *P. anomala* populations, performed using the UPGMA method based on the results of iPBS amplification.

The results of cluster analysis showed that the iPBS amplification results were positively correlated with the geographical data of the *P. anomala* sample collection site. As a result, a tree formed by three clades can be observed. The first clade is formed by populations P3 and P4, which are geographically very close. P1 and P2 of *P. anomala* populations that are also neighbor each other geographically form two mutually dependent branches on the dendrogram.

## 4. Discussion

In the study region, *P. anomala* grows under different ecological conditions and at altitudes (from 700 to 1850 m). Other authors [69–72] have confirmed the high ecological plasticity of the species. However, the species do not tolerate the excessive overwatering or drying of their habitat.

The morphological parameters of *P. anomala* study populations in East Kazakhstan (height, bush diameter, number of leaves per shoot) significantly exceed those of the Siberian populations. For example, the height of generative shoots in Siberia (in the Yamal-Nenets Autonomous Region) is 58.9 ± 2.5 cm [69], while the lowest individual height in the study populations of Eastern Kazakhstan is 78.6 ± 2.11 cm (P1). The average number of leaves per generative shoot in Siberia reached 5.6 ± 0.7 eq. [69], whereas in surveyed populations, the number of generative shoots is from 8.50 ± 0.32 eq. (P3) to 9.83 ± 0.28 eq. (P5). The morphometric parameters of *P. anomala* Kazakhstan populations are close to those of West Siberia, where adult generative individuals of the species have 3–17 shoots of 92–107 cm in height [70].

The analysis of the spectra of geographical elements of floras of various ranks, including the flora of plant communities within the scope of given classification units (cenoflora), is one of the main tools of comparative floristics [73]. The floristic composition of a plant community is a set of plant species that form communities of any rank and any type of vegetation [74]. From this viewpoint, the floristic composition of plant communities of species populations represents the association of historically and cenotic homogeneous groups of species within a syntax, which makes them the most important indicators of

plant cover from the level of a particular phytocenosis to altitudinal-belt divisions. These are the most important points of contact between floristics and geobotany [75].

A chorological analysis of the flora composition of *P. anomala* plant communities showed that the Eurasian (54%) and Asian groups (24%) were the most richly represented. Such a distribution of species ranges is generally characteristic of the flora of the study region. The systematic analysis of the species distribution of *P. anomala* communities also agrees with the data on the flora of Kazakhstan Altai [76].

Intensive land use, pollution, and the overexploitation of wild plant resources are causing a significant decline worldwide [77]. *P. anomala*, being a valuable botanical resource, underscores the essential need for the understanding of its biodiversity to preserve its bioresource potential. Current data indicate that research on this species has not been fully conducted, especially for the Kazakhstan populations of peonies, which are strongly affected by anthropogenic and anthropogenic factors, resulting in a greater risk of reducing reproduction and subsequent reduction in wild populations.

The analysis of the microscopic parameters of *P. anomala* leaves from different populations revealed that the maximum size of cells and tissues was reliably observed in P3, and the minimum in P2 and P1. No reliable differences between leaf plate lateral width and lower epidermis thickness were found. The general anatomical structures are the same; the populations differ by the presence or absence of trichomes. Populations P5, P4, and P1 have trichomes, and their lengths reliably differ according to the point of collection. In populations P3 and P2, trichomes were not found (Table 4).

When comparing the individual microscopic parameters of the *P. anomala* stem, it was found that there was no reliable difference between P5 and P4 populations in terms of epidermis thickness, chlorenchyma, and conductive bundle width. As for the previous leaf organ, in the stem structure, the maximum values of cell and tissue sizes were noted in the P3 population and the minimum was identified in P1 (Table 5).

For the leaf petiole, the scatter of dimensional characters for plant samples from different populations was also determined. No significant difference was found between the measures of epidermal thickness, most of the values of the thickness of chlorenchyma, or the thickness and length of the lateral conductive bundle. Maximum values of leaf petiole thickness were found in P3, and minimum values were found in P5. Maximum thickness of the epidermis was observed in P1, and the minimum in P5. Chlorenchyma thickness and lateral conductive bundle length were largest in specimens from P1 and smallest in P2. The maximum thickness of lateral conductive bundle and trichome length were recorded in plants from P1 and the minimum thickness was recorded in plants from P2 (Table 6).

A comparison of anatomical parameters of the leaf and stem in different populations showed a slight difference in the thickness of tissue sections and individual cells. The largest sizes of cells and tissues was observed in population P3, and the minimal values were observed in population P5.

Genetic diversity can be assessed using various modern approaches [70,78,79]. However, when selecting the optimal marker type, their prevalence in the genome and the possibility of identification via PCR methods play an important role. Retrotransposons, as a constantly changing composition of the genome of any eukaryotic species, have an advantage in the assessment of genetic polymorphism compared with other markers because they do not require prior data of the DNA sequence and give more reproducible results, which is especially important in studies of rare or little-studied plant species [78,80–82].

Retrotransposons are numerous, evolve rapidly, and are widespread in plant genomes. Retrotransposons are susceptible to environmental changes, so bursts of their activity can cause the induction of structural changes in the genome, contributing to rapid adaptation to changing external conditions, which is especially important for wild populations with low genetic diversity [83–85].

Many endemic species have low adaptive potential in changing climate conditions. Their adaptation is largely caused by the activation of retrotransposons and their ability to

regulate the expression of nearby genes in response to stress [86]. The dispersed nature of retrotransposons in the plant genome is one of the reasons why they can be used as a marker system for studying genetic polymorphism, just like AFLP and SSR markers [87].

Retrotransposons are unique, very stable, and can be considered Mendelian loci. They can be identified in various retrotransposon families via PCR amplification with primers derived from LTR (long terminal repeat) or conserved PBS sequences (primer binding sites). Various applications have now been developed to exploit polymorphisms in TE insertion patterns. Alternatively, retrotransposons can be mapped using high-performance next-generation sequencing and bioinformatics to develop primers that will yield simple-to-use codominant markers [88].

When using retrotransposons as markers, universal tRNA primer sites (iPBS—PrimerBindingSite) can be used to develop PCR primers [89]. The special mechanism of transposition inherent to retrotransposons, in which retrotransposon sequences are often located side-by-side in an inverted orientation, allows for amplification between the iPBS sections of retrotransposons using primers complementary to interspersed repeats [90].

This type of marker was used to study the genetic polymorphism of 65 tree peony specimens; it has been shown that iPBS can be used for the classification and research of the genetic diversity of various tree peony species [90]. With respect to populations of the shirking peony, studies to identify genetic polymorphisms have not been conducted; however, there are single publications using microsatellite markers [91] as well as chloroplast and nuclear genome markers [92]. The use of retrotransposons as markers for the analysis of *P. anomala* natural populations is mentioned only in one publication, where primers for the putative LTR sequences of retrotransposons were developed as a result of bioinformatics analysis. Due to the low efficiency of the developed primers, data on the analysis of *P. anomala* population polymorphism are not given [79].

The lowest indicators of biodiversity were found in the *P. anomala* population (P2), although this group was located in favorable conditions. This is probably because the area of projective covering of *P. anomala* in the given population was minimal (4–5%), owing to competition with other species, and due to the influence of anthropogenic factors, individuals of the given population were limited to interbreeding. One potential contributing factor is the significant amount of damage inflicted on *P. anomala* plants during logging activities. As a result, young *P. anomala* individuals are unable to recover from such damage. In addition, the mature and heavy seeds of *P. anomala* tend to disperse around the parent plant and are poorly transported by meltwater and wind due to dense grass vegetation. Distribution through endozoochry is not typical for these species; there are no reports of animals consuming its seeds [93].

Urbanization has had a significant effect on the natural environment, which leads to a decrease in the habitat of the species and impedes the flow of genes through seeds. Our research has revealed that the differentiation of the populations of Kazakhstan is relatively low, which may indicate the long-term existence of these populations as part of a larger population. Our assumption aligns with existing studies [94] that have established the rationale behind the low differentiation among populations of perennial *P. ludlowii*. This phenomenon is the result of genetic drift during a period when the species occupied adjacent habitats. Relatively low, but statistically significant, genetic differentiation among *P. ludlowii* populations is due to a shared ancestry rather than extensive gene flow.

The hierarchical arrangement of populations in the cluster further supports our assumption. The P5 population forms a separate branch, located at a distance from the other populations, which confirms the genetic isolation of this population. The geographic remoteness of the P5 population determines the accumulation of mutations in the gene pool, potentially giving rise to unique genotypes and genetic differentiation. In other populations, we observed the loss and fragmentation of habitat as a result of anthropogenic impacts, such as livestock grazing, deforestation, and the uncontrolled collection of *P. anomala* plants by the local community.

The low level of genetic differentiation of natural populations of *P. anomala* identified in this study underscores the significance of recognizing each population of *P. anomala* as a valuable genetic resource. The recognition of *P. anomala* as a valuable genetic resource is pivotal for developing a strategy for the conservation of this valuable medicinal plant. It is impossible to say that one population has greater or lesser priority for conservation than others; each holds equal importance in ensuring the survival of the species. Priority in developing a strategy for the conservation of genetic diversity should be given to the establishment of a germplasm resource bank for all populations of these species. This strategy should integrate in situ, ex situ, and in vitro conservation technologies to protect the genetic diversity of *P. anomala*.

We propose the existence of a single large population of *P. anomala* in the Kazakhstan Altai region on the basis of genetic polymorphism analysis using iPBS markers, the nature of population distribution, and genetic distances across populations. Despite the uniqueness of the amplification profiles in the P5 population, common loci characteristics of all *P. anomala* study populations were identified. This indicates genetic similarity and evolutionary unity among *P. anomala* populations.

**Author Contributions:** Conceptualization, S.A.K.; methodology, S.A.K., O.N.K. and M.Y.I.; software, A.S.T. and D.T.A.; validation, A.K.S., A.S.T., A.A.I. and M.Z.Z.; formal analysis, A.S.T.; investigation, S.A.K., A.S.T., O.N.K. and M.Y.I.; resources, S.A.K. and D.T.A.; data curation, S.A.K. and A.S.T.; writing—original draft preparation, S.A.K., A.S.T., O.N.K. and M.Y.I.; writing—review and editing, S.A.K.; visualization, S.A.K.; supervision, M.Z.Z.; project administration, M.Z.Z.; funding acquisition, S.A.K. All authors have read and agreed to the published version of the manuscript.

**Funding:** This research was funded by grants from the Ministry of Science and Higher Education of the Republic of Kazakhstan (No. AP19680461).

**Institutional Review Board Statement:** Not applicable.

**Informed Consent Statement:** Not applicable.

**Data Availability Statement:** The datasets analyzed in the present study are available from the author A.K. Sarkytbayeva upon request.

**Acknowledgments:** The authors are sincerely grateful to the anonymous reviewers for all their valuable comments on the first version of the manuscript.

**Conflicts of Interest:** The authors declare no competing interests.

## Appendix A. Floristic Composition of P. anomala Study Plots

| | SPECIES NAME | Abundance on the Brown–Blanke Scale (1928) | | | | | Ecological Group | Area |
|---|---|---|---|---|---|---|---|---|
| | | P1 | P2 | P3 | P4 | P5 | | |
| 1 | *Abies sibirica* Ledeb. | | | | | + | M | as. |
| 2 | *Achillea ledebourii* Heimerl | | 1 | | | 2 | M | as. |
| 3 | *Aconitum leucostomum* Vorosch. | | | 2 | 2 | 2 | MP | as. |
| 4 | *Aconitum septentrionale* Koelle | | 2 | | | | M | euras. |
| 5 | *Agrostis gigantea* Roth | | | | 3 | 3 | HM | holarc. |
| 6 | *Alfredia cernua* Cass. | | 1 | 1 | 2 | | HM | as. |
| 7 | *Allium microdictyon* Prokh. | | | | | 1 | HM | holarc. |
| 8 | *Alopecurus arundinaceus* Poir. | | 2 | | 1 | 2 | M | euras. |
| 9 | *Angelica sylvestris* L. | | 1 | | | | M | euras. |
| 10 | *Anthriscus sylvestris* (L.) Hoffm. | 2 | | 2 | 1 | 1 | M | euras. |
| 11 | *Aquilegia glandulosa* Fisch. ex Link. | | 1 | | | 3 | MX | as. |
| 12 | *Artemisia sericea* Weber ex Stechm. | 2 | | | | | M | euras. |
| 13 | *Artemisia vulgaris* L. | | | 2 | | | M | cosm. |
| 14 | *Betula pendula* Roth | | 1 | 1 | | + | M | euras. |
| 15 | *Bistorta officinalis* Delarbre | | | 2 | 1 | | M | euras. |

| | SPECIES NAME | Abundance on the Brown–Blanke Scale (1928) | | | | | Ecological Group | Area |
|---|---|---|---|---|---|---|---|---|
| | | P1 | P2 | P3 | P4 | P5 | | |
| 16 | *Brachypodium pinnatum* (L.) P.Beauv. | 1 | 2 | 1 | | 2 | M | euras. |
| 17 | *Bromus inermis* Leyss. | | 4 | | | | M | euras. |
| 18 | *Bupleurum aureum* Fisch. ex Hoffm. | 1 | | 2 | 1 | | M | as. |
| 19 | *Calamagrostis obtusata* Trin. | | 2 | 2 | | 1 | M | euras. |
| 20 | *Calamagrostis purpurea* (Trin.) Trin. | | 1 | | | 2 | HM | holarc. |
| 21 | *Campanula sibirica* L. | 2 | | | | | MX | euras. |
| 22 | *Caragana arborescens* Lam. | | 1 | | 1 | | MX | as. |
| 23 | *Carex acuta* L. | | 2 | | | | H | euras. |
| 24 | *Carex pediformis* var. *macroura* (Meinsh.) Kük. | | 1 | 1 | 1 | | MX | as. |
| 25 | *Cerastium arvense* L. | | 1 | | | 1 | M | holarc. |
| 26 | *Cirsium heterophyllum* (L.) Hill | 1 | | 1 | | 2 | M | euras. |
| 27 | *Clematis alpina* subsp. *sibirica* (L.) Kuntze | | 1 | 1 | | 1 | M | as. |
| 28 | *Clematis integrifolia* L. | | 1 | | | | MX | euras. |
| 29 | *Cotoneaster laxiflorus* J.Jacq. ex Lindl. | 2 | | | | | MX | euras. |
| 30 | *Cotoneaster uniflorus* Bunge | | | 1 | 1 | 2 | P | euras. |
| 31 | *Crepis capillaris* (L.) Wallr. | | 1 | 2 | | 1 | M | euras. |
| 32 | *Crepis sibirica* L. | | | | | 2 | M | euras. |
| 33 | *Dactylis glomerata* L. | 2 | 4 | 2 | 2 | 2 | M | euras. |
| 34 | *Delphinium elatum* L. | | | | | 2 | M | euras. |
| 35 | *Dracocephalum ruyschiana* L. | 1 | | 3 | | | M | euras. |
| 36 | *Elymus caninus* (L.) L. | 1 | 2 | | 2 | 1 | M | holarc. |
| 37 | *Equisetum sylvaticum* L. | | 1 | | | | M | holarc. |
| 38 | *Epilobium angustifolium* L. | | 1 | 5 | | 3 | M | holarc. |
| 39 | *Erythronium sibiricum* (Fisch. & C.A. Mey.) Krylov | | | | | 2 | M | as. |
| 40 | *Euphorbia pilosa* L. | | | | | 2 | MP | as. |
| 41 | *Festuca kryloviana* Reverd. | | | | | 2 | P | as. |
| 42 | *Filipendula ulmaria* (L.) Maxim. | | 3 | | | 2 | HM | euras. |
| 43 | *Filipendula vulgaris* Moench | 2 | | 3 | | | MX | euras. |
| 44 | *Fragaria vesca* L. | 3 | | 2 | | | M | cosm. |
| 45 | *Galatella sedifolia* (L.) Greuter | 1 | | 1 | 1 | | MX | as. |
| 46 | *Galium boreale* L. | 2 | 1 | 1 | 1 | 1 | M | holarc. |
| 47 | *Galium verum* L. | 2 | | 1 | | | M | cosm. |
| 48 | *Geranium albiflorum* Ledeb. | 1 | 1 | | | 2 | MP | as. |
| 49 | *Geranium albiflorum* Ledeb. | | 1 | | 2 | | MP | as. |
| 50 | *Geranium collinum* Stephan ex Willd. | 2 | | 2 | | 2 | MX | euras. |
| 51 | *Geranium pratense* L. | | | | 2 | | MX | euras. |
| 52 | *Geranium pseudosibiricum* Maxim. | 1 | | 1 | 1 | 1 | MX | euras. |
| 53 | *Geum rivale* L. | | | | | 1 | H | holarc. |
| 54 | *Hedysarum theinum* Krasnob. | | | | | 1 | MP | as. |
| 55 | *Heracleum dissectum* Ledeb. | | 1 | | | 1 | M | as. |
| 56 | *Heracleum sibiricum* L. | 1 | | 2 | 1 | | M | as. |
| 57 | *Hesperis matronalis* L. | | 1 | | | 1 | HM | as. |
| 58 | *Hypericum perforatum* L. | 2 | | 1 | | 1 | M | holarc. |
| 59 | *Iris ruthenica* Ker Gawl. | 1 | | | 1 | 1 | MX | as. |
| 60 | *Koenigia alpina* (All.) T.M. Schust. & Reveal | 1 | | 1 | | | M | euras. |
| 61 | *Lamium album* L. | 1 | | 2 | 1 | 1 | M | holarc. |
| 62 | *Larix sibirica* Ledeb. | | | | | 1 | M | euras. |
| 63 | *Lathyrus gmelinii* (Fisch. ex Ser.) Fritsch | 1 | 1 | | | 1 | HM | euras. |
| 64 | *Lathyrus pratensis* L. | 2 | | | 2 | | HM | euras. |
| 65 | *Leuzea carthamoides* (Willd.) DC. | | 1 | | | 1 | MP | as. |

| | Species Name | Abundance on the Brown–Blanke Scale (1928) | | | | | Ecological Group | Area |
|---|---|---|---|---|---|---|---|---|
| | | P1 | P2 | P3 | P4 | P5 | | |
| 66 | *Ligularia glauca* (L.) O.Hoffm. | 2 | 1 | | 2 | | MX | euras. |
| 67 | *Lilium martagon* L. | 1 | + | | | | M | euras. |
| 68 | *Linaria vulgaris* Mill. | 2 | 1 | 1 | | | M | euras. |
| 69 | *Lolium giganteum* (L.) Darbysh. | | | 1 | 1 | | M | euras. |
| 70 | *Lonicera caerulea* subsp. *altaica* (Pall.) Gladkova | 1 | | | | 1 | MP | as. |
| 71 | *Lonicera tatarica* L. | 1 | 1 | | + | | MX | euras. |
| 72 | *Melica altissima* L. | | 1 | | 2 | | MX | euras. |
| 73 | *Mentha asiatica* Boriss. | 1 | | | | | HM | holarc. |
| 74 | *Milium effusum* L. | | 1 | 1 | 1 | | M | holarc. |
| 75 | *Origanum vulgare* L. | | | 2 | | | M | euras. |
| 76 | *Oxalis acetosella* L. | | 1 | | | 1 | HM | holarc. |
| 77 | *Paeonia anomala* L. | 2 | 2 | 3 | 5 | 3 | M | as. |
| 78 | *Parasenecio hastatus* (L.) H.Koyama | 1 | 1 | | 1 | 1 | M | holarc. |
| 79 | *Peucedanum morisonii* Besser ex Schult. | | | 1 | | | MX | as. |
| 80 | *Phleum pratense* L. | | | | 3 | | M | euras. |
| 81 | *Phlomoides alpina* (Pall.) Adylov, Kamelin & Makhm. | | | | | 3 | P | as. |
| 82 | *Phlomoides tuberosa* (L.) Moench | 2 | | | | | MX | euras. |
| 83 | *Picea obovata* Ledeb. | | | | | + | HM | euras. |
| 84 | *Pinus sylvestris* L. | | | 1 | | | M | euras. |
| 85 | *Pleurospermum uralense* Hoffm. | | 1 | | 1 | 1 | M | euras. |
| 86 | *Poa pratensis* L. | 3 | 2 | 2 | 2 | 1 | M | holarc. |
| 87 | *Polemonium caeruleum* L. | 1 | 1 | 2 | 1 | 2 | M | euras. |
| 88 | *Populus tremula* L. | | 1 | | | | M | euras. |
| 89 | *Potentilla chrysantha* Trevir. | | 1 | 2 | 1 | | M | euras. |
| 90 | *Potentilla recta* L. | | | 2 | | | M | euras. |
| 91 | *Primula veris* subsp. *macrocalyx* (Bunge) Lüdi | 1 | 1 | 2 | 1 | | M | euras. |
| 92 | *Prunella vulgaris* L. | | 2 | | | | M | cosm. |
| 93 | *Prunus padus* L. | 1 | 1 | 1 | 1 | | M | euras. |
| 94 | *Pulmonaria mollis* Wulfen ex Hornem. | | 1 | | | 1 | M | euras. |
| 95 | *Ranunculus acris* L. | | | | | 2 | H | euras. |
| 96 | *Ranunculus repens* L. | 1 | | | | | H | euras. |
| 97 | *Ribes atropurpureum* C.A.Mey. | | | | | 1 | M | as. |
| 98 | *Ribes nigrum* L. | | | | | 2 | HM | euras. |
| 99 | *Rosa acicularis* Lindl. | 1 | 2 | | | 1 | M | holarc. |
| 100 | *Rosa spinosissima* L. | 4 | 1 | 4 | | | M | holarc. |
| 101 | *Rubus idaeus* L. | | 2 | | | 2 | M | euras. |
| 102 | *Rubus saxatilis* L. | | 2 | | | | M | euras. |
| 103 | *Rumex acetosella* L. | | | | 1 | | M | euras. |
| 104 | *Salix bebbiana* Sarg. | 1 | 1 | | | 1 | MP | euras. |
| 105 | *Salix caprea* L. | 1 | 1 | | | 1 | HM | as. |
| 106 | *Salix viminalis* L. | 1 | 1 | | | 1 | H | as. |
| 107 | *Sambucus racemosa* L. | | | | | 1 | M | euras. |
| 108 | *Sanguisorba officinalis* L. | 2 | 1 | 1 | 2 | 2 | M | holarc. |
| 109 | *Saussurea elegans* Ledeb. | | | 2 | 1 | | M | as. |
| 110 | *Saussurea latifolia* Ledeb. | | | | 3 | 5 | M | as. |
| 111 | *Serratula coronata* L. | | | 3 | | | M | euras. |
| 112 | *Silene graminifolia* Otth | 1 | | 1 | 2 | | P | as. |
| 113 | *Solidago virgaurea* L. | | 1 | | | 2 | M | euras. |
| 114 | *Sonchus arvensis* L. | 2 | | | | | M | cosm. |
| 115 | *Sorbus aucuparia* subsp. *glabrata* (Wimm. Grab.) Hedl. | 1 | | | | 1 | M | euras. |

| | Species Name | Abundance on the Brown–Blanke Scale (1928) | | | | | Ecological Group | Area |
|---|---|---|---|---|---|---|---|---|
| | | P1 | P2 | P3 | P4 | P5 | | |
| 116 | *Spiraea chamaedryfolia* L. | | 2 | 1 | | | M | euras. |
| 117 | *Spiraea media* Schmidt | 1 | | | 4 | 1 | M | euras. |
| 118 | *Stipa capillata* L. | 1 | | 1 | | | X | euras. |
| 119 | *Tanacetum vulgare* L. | 1 | | 1 | 2 | | M | euras. |
| 120 | *Thalictrum flavum* L. | | 1 | 1 | 1 | 2 | M | euras. |
| 121 | *Thalictrum flavum* L. | | 1 | 1 | 1 | 2 | M | euras. |
| 122 | *Trifolium pratense* L. | 1 | 2 | 3 | 1 | 1 | M | euras. |
| 123 | *Trollius altaicus* C.A. Mey. | | | | | 3 | MP | as. |
| 124 | *Urtica dioica* L. | 1 | 2 | | | 2 | M | cosm. |
| 125 | *Veratrum lobelianum* Bernh. | | 2 | 3 | 1 | 2 | MP | euras. |
| 126 | *Veronica longifolia* L. | | | | 2 | | M | euras. |
| 127 | *Vicia cracca* L. | 1 | 2 | 1 | 1 | 2 | M | holarc. |
| 128 | *Vicia sepium* L. | 2 | 1 | 1 | 2 | 2 | M | euras. |
| 129 | *Viola altaica* Ker Gawl. | | | | | 1 | P | as. |
| 130 | *Viola hastata* Michx. | 1 | | | | 1 | M | euras. |

Ecological groups: H—hygrophytes; HM- hygromesophytes; M—mesophytes; MX—mesoxerophytes; MP—mesopsychrophytes; X—xerophytes; P—psychrophytes. Area: The groups of floral elements: cosm.—Cosmopolitan; holarc.—Holarctic; euras.—Eurasian; as.—Asian.

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
