# Peer review of "Current State of Natural Populations of Paeonia anomala (Paeoniaceae) in East Kazakhstan"

_diversity, doi:10.3390/d15111127_

Round 1
Reviewer 1 Report
Comments and Suggestions for Authors
All comments to the manuscript are left as comments and marked in yellow.
The article submitted to me for review was, according to the title, supposed to deal with the genetic and intraspecific variability of Paeonia anomala.
Starting from the introduction, the content is not in line with the title and objectives. One gets the impression that everything (e.g. anatomy, habitat, chemical composition) is covered in this work, just not the genetic data. One gets the impression that everything was thrown in and the molecular data were given a cursory treatment. The quality of figures, citations, Latin names needs improvement.
The weakest part of this work is the choice of PBS markers. Especially as the authors suggest that several SSRs exist, so they could have been tested or new ones designed.
Bands analysis on a gel (whether PBS or AFLP) is not a very reproducible method and analysis of genetic parameters is severely limited. Hence the small number of results and very poor discussion of them.

Author Response
Author's Reply to the Review Report (Reviewer #1)
Reviewer #1 The article submitted to me for review was, according to the title, supposed to deal with the genetic and intraspecific variability of Paeonia anomala.
Starting from the introduction, the content is not in line with the title and objectives. One gets the impression that everything (e.g. anatomy, habitat, chemical composition) is covered in this work, just not the genetic data. One gets the impression that everything was thrown in and the molecular data were given a cursory treatment. The quality of figures, citations, Latin names needs improvement.
Authors’ response: The authors are grateful to the reviewer for their analysis of the manuscript. We concur with the reviewer's observation regarding discrepancies between the article's title and its content. Consequently, we have substantially revised the entire manuscript, including a complete overhaul of the figures and tables. The manuscript has been edited to incorporate all of the reviewer's comments and recommendations.
Reviewer #1 The weakest part of this work is the choice of PBS markers. Especially as the authors suggest that several SSRs exist, so they could have been tested or new ones designed.
Authors’ response:
The selection of PCR markers using PBS primers to identify genetic polymorphism is more effective than other existing PCR approaches, including simple sequence repeats (SSR) markers. This effectiveness arises from the benefits of multilocus genome analysis compared to single-locus analysis. Simultaneously analyzing numerous unlinked loci from different regions of chromosomes with just one primer simplifies and accelerates the assessment of genetic polymorphism and genetic diversity. We consider PBS-based markers, as well as SSR markers, as alternative methods for studying genetic polymorphism in various plant species, including Paeonia sp. This method is universal because interspersed repeats and transposable elements are present in all eukaryotic species, making it easily adaptable to any plant species, including Paeonia sp. The study of genetic polymorphism in P. anomala using SSR markers is less appealing to us, as we aim to avoid duplicating the work of other researchers. Additionally, developing new SSR markers is an expensive process, involving genome sequencing of several genetically distant genotypes and subsequent bioinformatics analysis to identify polymorphic loci, which is both challenging and costly. The use of such markers for wild plant populations is supported by the direct involvement of mobile elements in the adaptation of plants to stress. We believe that by using these markers, it is possible to identify genetic changes occurring in the genome due to exposure to various stress factors in wild populations of P. anomala living in the same region under different environmental conditions.
Reviewer #1 Bands analysis on a gel (whether PBS or AFLP) is not a very reproducible method and analysis of genetic parameters is severely limited. Hence the small number of results and very poor discussion of them.
Authors’ response: We conducted PBS polymorphism analyses, including amplification and DNA fingerprinting, twice for each DNA sample and have confidence in the reproducibility of this method. The method's reproducibility may be attributed to the standardized PCR reaction conditions in our country. If other authors adopt the exact protocol we have proposed, similar results should be achieved in their laboratories.
However, it's important to note that all PCR approaches are based on the RAPD method and may vary depending on the PCR conditions, which is a reasonable expectation. The unique characteristics of Taq polymerase and the high concentration of primers enable the RAPD method to function. Changing PCR conditions or using different polymerases, such as Phusion/Phire, can result in a completely different or similar spectrum of amplicons. However, within a specific protocol, results should be reproducible and effective.
A similar situation applies when using SSR markers. Altering the PCR conditions can lead to entirely unexpected results, such as the formation of a multilocus profile instead of a single band.
Reviewer #1 title to be changed. it is difficult to talk about genetic variation using PBS markers
Authors’ response: We changed the title of the article to "Current state of natural populations of Paeonia anomala L. (Paeoniaceae) in East Kazakhstan"
Lines 31-33: maybe the gene flow is behind it or vegetative reproduction
Authors’ response: Plant samples were collected from individuals situated at a minimum distance of 10 m. from each other, thus excluding the possibility of vegetative propagation.
Lines 49-50: notation of the Latin name; subsp and ssp
Author's response: we have rewritten this sentence as follows: The remaining species are reduced to synonyms.
Line 51: order of citations
Author's response: we fixed it
Lines 52-63: it adds nothing to the text and is not in keeping with the title
Author's response: We believe general information about the beneficial properties of the research object is important, for the readers.
Lines 85-88: SSR markers seem to be the best for population studies. If such markers have been used for other Paeonia species or primers are available, why have they not been used for cross-amplification?
Authors’ response: The SSR method, like any other PCR method, is sensitive to changes in reaction conditions, which can affect the expected results. Implementing the SSR method requires considerable effort and testing to select the most effective markers for professional use. While the RAPD-related PCR fingerprinting methods we described above are straightforward, they do present a significant challenge when it comes to electrophoretic separation of multilocus amplicons.
Our study contributes new insights into the genetic polymorphism of wild populations of P. anomala in Eastern Kazakhstan. We have also highlighted the strengths and weaknesses of alternative analysis methods to underscore the potential of our proposed method for estimating the biodiversity of P. anomala.
Lines 92-95: a mere exaggeration. maybe for phylogenetic studies they are good but not for population studies
Authors’ response: A number of studies are available that favor multilocus analysis (e.g., RAPD, ISSR, iPBS, etc.) rather than single-locus analysis (e.g., SSR). This type of markers, along with SSR markers, is specifically used for the analysis of population polymorphism, and in several studies, it has been shown to be equally effective in terms of resolution. For example, refer to the following studies:
- Kalendar, R., Amenov, A., & Daniyarov, A. (2018). Use of retrotransposon-derived genetic markers to analyse genomic variability in plants. Functional Plant Biology. doi:10.1071/fp18098.
- de Carvalho Santos, T.T., de Oliveira Amorim, V.B., dos Santos-Serejo, J.A. et al. Genetic variability among autotetraploid populations of banana plants derived from wild diploids through chromosome doubling using SSR and molecular markers based on retrotransposons. Mol Breeding 39, 95 (2019). https://doi.org/10.1007/s11032-019-0996-1
- Razi, M., Amiri, M.E., Darvishzadeh, R. et al. Assessment of genetic diversity of cultivated and wild Iranian grape germplasm using retrotransposon-microsatellite amplified polymorphism (REMAP) markers and pomological traits. Mol Biol Rep 47, 7593–7606 (2020). https://doi.org/10.1007/s11033-020-05827-3
- Li, S.; Ramakrishnan, M.; Vinod, K.K.; Kalendar, R.; Yrjälä, K.; Zhou, M. Development and Deployment of High-Throughput Retrotransposon-Based Markers Reveal Genetic Diversity and Population Structure of Asian Bamboo. Forests 2020, 11, 31. https://doi.org/10.3390/f11010031
Line 106: po prostu morfologia lub morfometria, co to jest biomorfologia?
Author's response: We have corrected the term "bimorphology" to the term "morphology"
Lines 196-197: this should have been mentioned in the introduction. this has a very strong impact on species biology and the interpretation of genetic data
Author's response: We have added one sentence in the introduction: Seed regeneration of peony in natural habitats is usually low, as seeds are damaged by weevil larvae and have delayed seed germination and seedling formation due to weak embryo development and low activity of basic enzymes [38].
Lines 236-237: nothing can be seen in the figure. which characters correlates with which? in this case the correlation analysis adds nothing. it is clear that morphological traits are strongly influenced by environment and location (phenotypic plasticity) and that there is a very strong correlation between e.g. height and stem diameter
Author's response: We agree with this comment; we have removed the figures of correlation analyses from the text of the paper.
Lines 262: biomorphology (highlighted)
Author's response: We changed "Biomorphology" to "Morphology."
Lines 274: necessarily to the supplement
Author's response: We have moved this table to the Appendix
Lines 297-298: quality of Figures 5, 6 and 7, and in my view they add nothing to the text
Author's response: We have removed these drawings from the article
Lines 309-312: italic
Author's response: We've corrected the text to italicize
Lines 322-324: and what about the repeatability of the bands? was there a repeated PCR to compare the applicon patterns?
Authors’ response: To ensure repeatability, we conducted all PBS polymorphism analyses (including amplification and DNA fingerprinting) twice for each DNA sample. We can confidently assert the reproducibility of this method when our developed protocol is strictly adhered to.
Lines 365: order of subsections, this should come before genetic testing
Author's response: We have moved this section above
Lines 366-377: what do the anatomical findings contribute to the tect and objectives? not in accordance with the title
Author's response: intraspecific variability of plants implies changes in indicators not only at the population level, variation of genes, morphological parameters, but also anatomical structures. Morphological and anatomical indices in general reflect quite well the degree of development of individuals in certain soil and climatic conditions. These indicators can become the basis for monitoring the state of species populations.
Lines 516-518: please convince me that the results are reproducible!!!
Authors’ response: We have already addressed this question above. Furthermore, in order to confirm the repeatability of our results, we conducted all PBS polymorphism analyses (including amplification and DNA fingerprinting) twice for each DNA sample. We confidently affirm the reproducibility of this method. Additionally, the reproducibility of the methods can also be indirectly determined by:
- The presence of common bands among genetically distinct samples.
- The existence of polymorphic bands that are also replicated in other genetically distinct samples.
- The spectra revealing a common structure for closely related samples, where samples from the same population or closely related ones exhibit characteristic shared spectra, and similar samples are grouped together.
- Genetically distant samples will exhibit a greater number of differing bands between them
Lines 526-531: The fact that PBS markers were only used in 1 publication is not sufficient reason to use them or that this is the purpose of the study. Microsatellites in this case would be better.
Authors’ response: We have previously discussed the strengths and weaknesses of various analysis methods, including SSR markers. Our study illustrates the potential of our proposed multilocus analysis method for assessing the diversity of species that face extinction or environmental stress. We believe that this method deserves a place alongside other multilocus analysis methods (such as RAPD, ISSR, iPBS, etc.) and single-locus analysis (SSR). These types of markers, including SSR markers, are commonly used for analyzing population polymorphism in numerous plant species, and in several studies, they have demonstrated comparable resolution to existing methods.
Lines 535-545: poor statistical parameters, their interpretation can be cumbersome and this is due to inadequate markers
Authors’ response: We acknowledge that the genetic diversity rates are low. We are confident that by using Paeonia sp. specimens that are more genetically and geographically distant, these figures would be higher. A low level of genetic diversity is a common trait in many wild Paeonia species, influenced by various factors, such as geographic isolation due to the Altai mountain system and adaptation to specific habitat conditions.
In our case, we postulate that the low level of diversity may also be attributed to the fact that all populations of Paeonia anomala are located within the same geographical area. Additionally, Paeonia anomala holds significant value as a medicinal plant, utilized in both official and folk medicine. The populations of this species are under substantial pressure from anthropogenic factors, with uncontrolled mass collection occurring in their natural habitats.
Our results align with studies on other peony species:
- Y.J. Zhao, G.-S. Yin, X. Gong RAD-sequencing improves the genetic characterization of a threatened tree peony (Paeonia ludlowii) endemic to China: Implications for conservation, Plant Diversity, 2022. https://doi.org/10.1016/j.pld.2022.07.002.
- Wang, SQ. Genetic diversity and population structure of the endangered species Paeonia decomposita endemic to China and implications for its conservation. BMC Plant Biol 20, 510 (2020). https://doi.org/10.1186/s12870-020-02682-z
- Xu, X.-X., Cheng, F.-Y., Xian, H.-L., & Peng, L.-P. (2016). Genetic diversity and population structure of endangered endemic Paeonia jishanensis in China and conservation implications. Biochemical Systematics and Ecology, 66, 319–325. doi:10.1016/j.bse.2016.05.003
In addition, the assessment of the probability of error carried out by the permutation method confirmed the reliability of differences between populations (the index of genetic differentiation ɸPT=0.364, which exceeds P < 0.001).
Lines 535-545: or maybe it is due to vegetative reproduction, so we have clonality and low variability. Low gene flow because such large and heavy seeds have poor dispersal. Or maybe isolation by IBD distance works.
Authors’ response: Your valuable point warrants further discussion. We collected samples from individuals growing at a considerable distance from each other, thus excluding the possibility of vegetative propagation. We concur with the reviewer that the size of the seeds makes them challenging to disperse. However, it's important to consider that the projective cover of P. anomala (the area projected by the species per unit area) is very low. The dense grass cover of other species limits hybridization. Consequently, P. anomala struggles to compete with other species in this area.
Moreover, mature seeds, being heavy, fall in proximity to the mother plant and are poorly transported by wind and meltwater, and they are not typically consumed by animals and birds. Additionally, anthropogenic factors, such as collection by the local population, animal grazing, and deforestation, further impact these plants. We have included this information in the discussion (the final version is provided below).
Lines 534-553 The lowest indicators of biodiversity were found in P. anomala population (P2), despite the fact that it is located in favourable conditions of existence. Probably, it is because the area of projective covering of P. anomala in the given population is minimal (4-5 %), owing to low competition with other species, and also the influence of the anthropogenic factor, individuals of the given population are limited in the interbreeding. We assume that one of the reasons for this is the substantial damage inflicted upon P. anomala plants during logging. Consequently, young P. anomala individuals struggle to recover from such damage. Furthermore, the mature and heavy seeds of P. anomala fall in proximity to the mother plant and are poorly transported by meltwater and wind due to the dense grass stand. This species does not typically rely on endozoochory for distribution, and there are no reports of animals consuming it (Ahmad, Tabassum, 2013).Urbanization also significantly affects the habitat, leading to a reduction in the species' habitat and hindering gene flow through seeds. Our studies have indicated that the differentiation among the populations in Kazakhstan is low, which may suggest their long-term existence within a single large population. Our assumption aligns with Y-J's studies, specifically Zhao et al. (2022), who discovered that the low genetic differentiation among perennial P. ludlowii populations results from genetic drift during the period when the species' habitat was contiguous (Zhao et al. (2022). The low, yet statistically significant, genetic differentiation among P. ludlowii populations is attributed more to a common ancestry than gene flow The hierarchical arrangement of populations in the cluster confirmed this assumption. A separate branch is formed by the P5 population, located remotely from the other populations, which confirms the genetic isolation of this population. The geographic remoteness of this population determines the accumulation of mutations in the gene pool, which may contribute to the appearance of unique genotypes and genetic differentiation. In contrast, in other populations, we observe habitat loss and fragmentation as a result of anthropogenic impacts, including livestock grazing, deforestation, and the uncontrolled collection of P. anomala plants by the local population.
The low level of genetic differentiation among natural populations of P. anomala identified in the present study suggests that each population of P. anomala represents a valuable genetic resource. This fact must be taken into account when developing conservation strategies for this valuable medicinal plant. It would be unwise to argue that any one population has more or less priority for conservation compared to others, as each population is crucial for the species' survival. Therefore, when formulating strategies for preserving genetic diversity, the top priority should be the establishment of a germplasm resource bank that includes representatives from all populations of the species, utilizing in situ, ex situ, and in vitro conservation technologies.
Lines 554: I would divide the methodology into sections, according to the results
Authors’ response: We moved the methodology section above after the introduction and divided according to the results obtained: 2. Materials and methods - 2.1 Ecological, biological and floristic studies of P. anomala populations; 2.2 Study of genetic variability of P. anomala; 2.3 Methods for investigating the anatomical structure of P. anomala
Lines 572: ???
The life cycles of peonies were studied according to the method of Uranov A.A.

Reviewer 2 Report
Comments and Suggestions for Authors
The abstract, introduction, and methodology is well written.
1. While the article mentions the satisfactory condition of P. anomala populations in the study region, it would be helpful to provide more specific information or data on population size, distribution, and any observed trends over time. This would provide a clearer understanding of the species' conservation status.
2. The article mentions the floristic composition of P. anomala plant communities, but it would be valuable to include information on the ecological characteristics of these communities, such as soil types, microclimate, and associated plant species. These details could help explain the species' preference for specific habitats.
3. The article briefly mentions the genetic differentiation of Kazakhstan populations but does not provide a clear explanation of the implications or significance of this finding. Further discussion on the potential factors influencing genetic differentiation and its conservation implications would enhance the understanding of the species' population dynamics.
Comments on the Quality of English LanguageThe English is fine.
Author Response
Author's Reply to the Review Report (Reviewer #2)
The abstract, introduction, and methodology is well written.
- While the article mentions the satisfactory condition of anomala populations in the study region, it would be helpful to provide more specific information or data on population size, distribution, and any observed trends over time. This would provide a clearer understanding of the species' conservation status.
Authors’ response: we thank to review for their work done and the great help provided to the authors in the work on the manuscript. Unfortunately, earlier on the territory of Kazakhstan, studies on the population size and distribution of P. anomala have not been conducted. For this reason, we cannot trace the dynamics of changes in the number and spatial distribution of the species. However, we believe that our study may become a starting point for further studies on the status of P. anomala populations.
- The article mentions the floristic composition of P. anomala plant communities, but it would be valuable to include information on the ecological characteristics of these communities, such as soil types, microclimate, and associated plant species. These details could help explain the species' preference for specific habitats.
Authors’ response: This information is presented in the revised version of the article – Lines 225-361.
- The article briefly mentions the genetic differentiation of Kazakhstan populations but does not provide a clear explanation of the implications or significance of this finding. Further discussion on the potential factors influencing genetic differentiation and its conservation implications would enhance the understanding of the species' population dynamics.
Authors’ response: We thank the reviewer for valuable comments. This study marks the first exploration of the genetic diversity within Kazakhstani populations of P. anomala, incorporating both morphological characteristics and molecular markers. Our research has unveiled that despite the limited differentiation among natural P. anomala populations, each one represents a valuable genetic resource. This revelation underscores the importance of considering these populations when devising conservation strategies for this invaluable medicinal plant.
A priority in this endeavor is the creation of a germplasm resource bank for this species, employing in situ, ex situ, and in vitro conservation technologies. We have included information stressing the necessity of preserving the biodiversity of this species in our revised article.

Reviewer 3 Report
Comments and Suggestions for Authors
The manuscript presents interesting topic, it characterizes endangered peony species from different points of view. The major part of the manuscript deals with morphologic and ecologic characteristic, minor part represents describing genetic variation using iPBS markers. Data have potential for publication in the Diversity, but manuscript contains several issues to be improved before it is accepted.
MAJOR ISSUES
1) Manuscript has very unusual (and not logically constructed) structure. Section Material & Methods is in the Diversity journal the 2nd chapter/section, as it usual in vast majority of scientific papers. In this manuscript is without any logic placed after the discussion. Some parts of Results belong to Material and Methods (for instance Table 4). Besides the Results, other parts of manuscript are not structured, I really recommend to divide text to subchapters for better understanding.
2) Results of Genetic diversity is poorly presented. Authors provide unnecessary image of fragmentation analysis on the gel, but table of scored alleles (for instance in supplementary data, not included in the manuscript) would be much more informative. Reader has no idea on what data are based presented population genetic characteristics. The lack of elementary data for genetic diversity is in contrary to very extensive morphologic and ecologic data.
I recommend to improve and much better explain methodology and results of the “genetic diversity” part and reduce the large part of morphologic and ecologic data – I recommend to summarize data and place some data to supplementary data (especially Table 3, which interrupts the text and information is too detailed for the main body of the manuscript).
I strongly suggest authors to precisely describe the plant material used in the study. It is very confusing how many plamnts were used for each part of the study and it is not directly clear how many plants were used for genotyping (and author should not hide this information and leave to reader to just guess how many plants were tested). This makes the paper a bit unreliable.
If the paper is better structured and data on morphology/ecology and genetic diversity are more balanced, I believe it might be worth publishing.
MINOR ISSUES
TITLE
Major part of the paper consist of presenting data on morphology and ecologic characteristic of peony population in Kazakhstan, genetic diversity is really a minor part. Therefore the title does not correspond to content of the paper. Title must changed according to its major outcomes.
LIST OF AUTHORS
Line 5 — Somebody is missing at the end of list of authors? Address 7 is also missing among consequent affiliations.
Acronyms/Abbreviations/Initialisms should be defined the first time they appear in each of three sections: the abstract; the main text; the first figure or table. When defined for the first time, the acronym/abbreviation/initialism should be added in parentheses after the written-out form.
--> abbreviations should be explained according to the Instructions for Authors. iPBS, CTAB, CIS countries,
TAXONOMIC/NOMENCLATORIAL ISSUES
Scientific names are written italics, but taxon ranks (var., subsp.) are written in regular font: incorrectly applied in Table 3, rows: 24, 28, 70, 91, 115
Wrong authorship of taxa:
In the text: page 1,
In Table 3:
row 39 - correctly (Fisch. & C.A. Mey.) Krylov
row 48 – “Ledeb.” Incorrectly in italics
row 60 – correctly (All.) T.M. Schust. & Reveal
row 81 - correctly (Pall.) Adylov, Kamelin & Makhm.
Lines 43-44 – name Paeonia hybrid is not correct name for the taxon.
lines 45-46 missing or incorrect authorship (“CA” should be C.A., “KY” should be “K.Y.”. Name of taxon Paeonia sinjiangensis is incorrectly written as Paeonia sinjangensis. Authorship (C.A. Mey. And K.Y. Pan) should be written in regular font, not in italics. Same mistake is on next page, line 48.
Line 49-50 – confusing name of taxon “P. anomala subsp. Anomala ssp. Veitchii (Lynch) DY Hong KY Pan” – correct taxon name at the subspecies level is Paeonia anomala subsp. veitchii (Lynch) D.Y. Hong & K.Y. Pan
Wrong names of taxa:
Authors stated that nomenclature follows POWO, but names of taxa are used inconsistently:
Lonicera altaica Pall. (line 115) versus L. coerulea subsp. altaica (Table 3, row 70)
Chamaenerion angustifolium (line 191) versus Epilobium angustifolium (Table 3, row 26)
Typing errors in taxa names:
Trifolium pretense (line 310)
Scientific names of taxa not written italics (line 30í-313)
Trollius Altaicus (line 194)
TYPING ERRORS
Many typing errors throughout whole manuscript, just for example errors on page 2:
- delete space before dot (line 59)
- change double space to single space (line 56, 67)
- add space after comma (line 51, 59, 62, 84)
- add space before parentheses (line 59, 63)
OTHER ISSUES
line 75-76: Does the Red Book of Kazakhstan list species in international recognized (IUCN) categories of threat? If so, authors should mention to which category of threat (conservation status) it belongs. (“Rare” is not any category, should be LC, NT, VU, EN or CR)
Page 5, Figure 2 – scale is nearly invisible, small map of Kazakhstan is for non-Kazakhstani readers uneasy to understand: it is unclear what is the Kazakhstan in the map, neighbouring countries are named in Russian. I recommend to highlight borders of Kazakhstan and used either international abbreviation of countries, or name them in English).
Unify the name of DNA markers used, I recommend to use the name mentioned in the original paper by Kalendar et al. 2010, i.e. the “iPBS” (see lines 347 and 525) instead of just “PBS” (all other mentions in the manuscript, i.e. lines: 30,35, 327, 329, 337, 352, 358, 359, 522, 548, 586)
line 579 – „… lysis STAB buffer“ — Do authors mean the CTAB buffer? If so, I recommend to correct it to „… CTAB (cetrimoniumbromid) extraction buffer“. I also recommend adding appropriate citation of the protocol used (either the classic paper of Doyle & Doyle 1987, or citation of some modified protocol that authors used for DNA extraction)
line 586 – marker should be explained at first mention in Methods section. I recommend changing “Universal PBS primers were applied to estimate the genetic diversity of P. anomala populations [52].” to “Genetic diversity of the P. anomala populations was estimated by using the Inter-primer binding site (iPBS) retrotransposon markers [52]”.
Line 628 – I do not know the MDPI policy, however I recommend to clearly state the contribution of each author. Statement “All the authors worked and wrote the manuscript.” sounds uncertain (and untrustworthy).
Line 777 – wrong citation! – author of cited paper is not Ruslan Kalendar only, but O. Khapilina and other 6 authors (including R. Kalendar).
Comments on the Quality of English LanguageENGLISH
English of the manuscript is often using inappropriate terms and repeatedly it is very difficult to follow. Examples of inappropriate terms are:
- line 73 – not spreading, but distribution
- line 297 – not “leading families”, but “major families” or “the commonest families”
Example of sentence, which requires considerable rewriting is on lines 315-316 (“amplification of P. anomala DNA resulted in clearly distinguishable amplicons, varied in number”) [Did authors mean “Amplified PCR products varied in number among different primers, or among samples, or what?]. Or lines 72-75 (“hearts of spreading” [Did authors mean “centre of distribution”?], “attribute to less extensive areas of distribution”). And unfortunately, throughout the whole manuscript.
Above mentioned examples are just few of many I strongly suggest to submit paper to some professional Language Editing service.
Author Response
Author's Reply to the Review Report (Reviewer #3)
The manuscript presents an interesting topic, characterizing endangered peony species from various perspectives. Most of the manuscript is dedicated to morphological and ecological characteristics, with a smaller section describing genetic variations using iPBS markers. The data has the potential to be published in the journal Diversity, but the manuscript does contain several issues that require improvement before it can be accepted.
Author's response:
We thank to review for the work done and for the great help provided to the authors in the work on the manuscript.
MAJOR ISSUES
Reviewer #1: Manuscript has very unusual (and not logically constructed) structure. Section Material & Methods is in the Diversity journal the 2nd chapter/section, as it usual in vast majority of scientific papers. In this manuscript is without any logic placed after the discussion. Some parts of Results belong to Material and Methods (for instance Table 4). Besides the Results, other parts of manuscript are not structured, I really recommend to divide text to subchapters for better understanding.
Authors’ response: We have restructured the manuscript based on the reviewer's comments and recommendations. The Materials and Methods section has been moved to Section 2, and Table 4 has been relocated to the Materials and Methods section. We have improved the overall structure of the manuscript. 2. Materials and methods - 2.1 Ecological, biological and floristic studies of P. anomala populations; 2.2 Study of genetic variability of P. anomala; 2.3 Methods for investigating the anatomical structure of P. anomala
Reviewer #2 Results of Genetic diversity is poorly presented. Authors provide unnecessary image of fragmentation analysis on the gel, but table of scored alleles (for instance in supplementary data, not included in the manuscript) would be much more informative.
Authors’ response: We employ the standardized iPBS method protocol, which was initially described in detail in the article.
Kalendar R, Antonius K, Smykal P, Schulman AH 2010. iPBS: A universal method for DNA fingerprinting and retrotransposon isolation. Theoretical and Applied Genetics, 121(8): 1419-1430. DOI:10.1007/s00122-010-1398-2.
После в работе Kalendar R, Schulman AH 2014. Transposon based tagging: IRAP, REMAP, and iPBS. Methods in Molecular Biology, 1115: 233-255. DOI:10.1007/978-1-62703-767-9_12
A detailed protocol has been published.
We have observed that many researchers use this method with protocol violations, particularly during the PCR and DNA fingerprinting stages. Consequently, results are often misinterpreted, gel quality suffers, and in some cases, gel images are entirely missing. Such issues discredit the effectiveness of this multilocus genome analysis method, which offers resolution no less than widely-used methods like SSR and ISSR.
In light of this, we have found it necessary to provide a more comprehensive and detailed explanation of this method. Our intention is to help fellow researchers improve the quality of their work in this area.
If other authors follow our proposed protocol exactly, then similar and reproducible results can be achieved in their laboratories.
Reviewer #3 Reader has no idea on what data are based presented population genetic characteristics. The lack of elementary data for genetic diversity is in contrary to very extensive morphologic and ecologic data.
I recommend to improve and much better explain methodology and results of the “genetic diversity” part and reduce the large part of morphologic and ecologic data – I recommend to summarize data and place some data to supplementary data (especially Table 3, which interrupts the text and information is too detailed for the main body of the manuscript).
Authors’ response: We have essentially rewritten the entire manuscript and revised the figures and tables. The manuscript has been thoroughly edited to address all of the reviewer's comments and recommendations. Additionally, we are including the results of the genetic analysis of P. anomala populations.
Reviewer #3 I recommend to improve and much better explain methodology and results of the “genetic diversity” part and reduce the large part of morphologic and ecologic data – I recommend to summarize data and place some data to supplementary data (especially Table 3, which interrupts the text and information is too detailed for the main body of the manuscript).I strongly suggest authors to precisely describe the plant material used in the study. It is very confusing how many plamnts were used for each part of the study and it is not directly clear how many plants were used for genotyping (and author should not hide this information and leave to reader to just guess how many plants were tested). This makes the paper a bit unreliable.
Authors’ response: We have adjusted the text in the Materials and Methods section to specify the sample size for each study, as per the reviewer's recommendations.
Reviewer #3: If the paper is better structured, with a more balanced presentation of morphology, ecology, and genetic diversity data, I believe it could be worth publishing.
Authors’ response: We have restructured the entire article following the reviewer's suggestions, and essentially, we have rewritten the entire manuscript
MINOR ISSUES
TITLE
Major part of the paper consist of presenting data on morphology and ecologic characteristic of peony population in Kazakhstan, genetic diversity is really a minor part. Therefore the title does not correspond to content of the paper. Title must changed according to its major outcomes.
Authors’ response: We changed the title of the article to "Current state of natural populations of Paeonia anomala L. (Paeoniaceae) in East Kazakhstan"
LIST OF AUTHORS
Line 5 — Somebody is missing at the end of list of authors? Address 7 is also missing among consequent affiliations.
Authors’ response: We have corrected the list of authors and affiliations
Acronyms/Abbreviations/Initialisms should be defined the first time they appear in each of three sections: the abstract; the main text; the first figure or table. When defined for the first time, the acronym/abbreviation/initialism should be added in parentheses after the written-out form. abbreviations should be explained according to the Instructions for Authors. iPBS, CTAB, CIS countries,
Authors’ response: We have made the appropriate edits to the text of the article
TAXONOMIC/NOMENCLATORIAL ISSUES
Scientific names are written italics, but taxon ranks (var., subsp.) are written in regular font: incorrectly applied in Table 3, rows: 24, 28, 70, 91, 115
Wrong authorship of taxa:
In the text: page 1,
In Table 3:
row 39 - correctly (Fisch. & C.A. Mey.) Krylov
row 48 – “Ledeb.” Incorrectly in italics
row 60 – correctly (All.) T.M. Schust. & Reveal
row 81 - correctly (Pall.) Adylov, Kamelin & Makhm.
Lines 43-44 – name Paeonia hybrid is not correct name for the taxon.
lines 45-46 missing or incorrect authorship (“CA” should be C.A., “KY” should be “K.Y.”. Name of taxon Paeonia sinjiangensis is incorrectly written as Paeonia sinjangensis. Authorship (C.A. Mey. And K.Y. Pan) should be written in regular font, not in italics. Same mistake is on next page, line 48.
Line 49-50 – confusing name of taxon “P. anomala subsp. Anomala ssp. Veitchii (Lynch) DY Hong KY Pan” – correct taxon name at the subspecies level is Paeonia anomala subsp. veitchii (Lynch) D.Y. Hong & K.Y. Pan
Wrong names of taxa:
Authors stated that nomenclature follows POWO, but names of taxa are used inconsistently:
Lonicera altaica Pall. (line 115) versus L. coerulea subsp. altaica (Table 3, row 70)
Chamaenerion angustifolium (line 191) versus Epilobium angustifolium (Table 3, row 26)
Typing errors in taxa names:
Trifolium pretense (line 310)
Scientific names of taxa not written italics (line 30í-313)
Trollius Altaicus (line 194)
Authors’ response: We have corrected all taxonomic and nomenclatural errors in the text
TYPING ERRORS
Many typing errors throughout whole manuscript, just for example errors on page 2:
- delete space before dot (line 59)
- change double space to single space (line 56, 67)
- add space after comma (line 51, 59, 62, 84)
- add space before parentheses (line 59, 63)
Authors’ response: We've fixed all the typos in the text
OTHER ISSUES
line 75-76: Does the Red Book of Kazakhstan list species in international recognized (IUCN) categories of threat? If so, authors should mention to which category of threat (conservation status) it belongs. (“Rare” is not any category, should be LC, NT, VU, EN or CR)
Authors’ response: P. anomala is not included in the IUCN International Red List. This species is protected only at the regional level (in Kazakhstan), as it is intensively harvested as a medicinal raw material.
Page 5, Figure 2 – scale is nearly invisible, small map of Kazakhstan is for non-Kazakhstani readers uneasy to understand: it is unclear what is the Kazakhstan in the map, neighbouring countries are named in Russian. I recommend to highlight borders of Kazakhstan and used either international abbreviation of countries, or name them in English).
Authors’ response: We fixed the map, added international abbreviations of neighboring countries
Unify the name of DNA markers used, I recommend to use the name mentioned in the original paper by Kalendar et al. 2010, i.e. the “iPBS” (see lines 347 and 525) instead of just “PBS” (all other mentions in the manuscript, i.e. lines: 30,35, 327, 329, 337, 352, 358, 359, 522, 548, 586)
Authors’ response: The text has been amended accordingly
line 579 – „… lysis STAB buffer“ — Do authors mean the CTAB buffer? If so, I recommend to correct it to „… CTAB (cetrimoniumbromid) extraction buffer“. I also recommend adding appropriate citation of the protocol used (either the classic paper of Doyle & Doyle 1987, or citation of some modified protocol that authors used for DNA extraction)
Authors' response: Appropriate changes were made to the corresponding section of the manuscript.
Visualization of extracted DNA was performed in 1% agarose gel by using the PharosFXPlus (Bio-RadLaboratories Inc., США). DNA samples were prepared in two variants: uterine solution for long- term storage at -20°C; working solutions were used for PCR at a concentration of 10 ng/μl.
Universal PBS primers were applied to estimate the genetic diversity of P. anomala populations [52]. The sequences of the primers used, which are complementary to the sites of different retrotransposons, are presented in Table 5. The PCR reaction was performed in a volume of 20 μl of reaction mixture including 3 μl DNA (10 ng/μl), 4 μl Phire Reaction Buffer 5x with MgCl 2, 1 μl primer (10 mM), 0.4 μl dNTPs mixture (10 mM), 0.2 μl 1 U Phire Hot Start polymerase. The mode of amplification was as follows: pre-denaturation at 98°C for 2 min, then 30 cycles: 98°C for 30 s, 50- 57°C for 1 min, 72°C for 1 min, and additional elongation at 72°C for 2 min. Amplification was performed in a SimpliAmp™ amplifier (ThermoFisherScientific Inc., USA). All PCRs were repeated at least twice for each DNA samples. PCR products were visualized in 1.5% agarose gel with ethidium bromide added. The sizes of amplified DNA fragments were determined by comparing them with a marker (Thermo Scientific GeneRuler DNA Ladder Mix 100-10,000 bp). Fragment lengths were determined using the Quantity One program in the PharosFXPlus gel documentation system (Bio- RadLaboratories Inc., USA).
For DNA fingerprinting, only clear, scorable bands were considered. Each uniquely sized band corresponds to a unique locus. When constructing a binary matrix, reproduced fragments were scored as either present (1) or absent (0)The level of detectable polymorphism was determined by the percentage of polymorphic amplicons to the total number of amplicons for each primer. asic indicators of genetic biodiversity, such as the number of alleles, Shannon information index (I), and the index of genetic differentiation (PhiPT), were determined using GenAlEx 6.5
The dendrogram was constructed using the UPGMA method [52].
line 586 – marker should be explained at first mention in Methods section. I recommend changing “Universal PBS primers were applied to estimate the genetic diversity of P. anomala populations [52].” to “Genetic diversity of the P. anomala populations was estimated by using the Inter-primer binding site (iPBS) retrotransposon markers [52]”.
Authors’ response: Appropriate changes were made to the corresponding section of the manuscript.
Line 628 – I do not know the MDPI policy, however I recommend to clearly state the contribution of each author. Statement “All the authors worked and wrote the manuscript.” sounds uncertain (and untrustworthy).
Authors’ response: Added to the text: Contribution of the authors: Conceptualization, S.A.K.; methodology, S.A.K., O.N.K. and M.Y.I.; formal analysis, A.K.S., A.S.T., A.A.I., A.B.M., M.Z.Z.; study, P. anomala ecologi-cal and phyto-cenotic confinement and morphology, S.A.K., A.K.S., D.T.A., A.B.M; Variability in the anatomical structure of P. anomala aboveground organs, M.Y.I.; Genetic diversity of P. anomala population, O.N.K., A.S.T.; Data curation, S.A.K, Initial draft preparation, S.A.K., O.N.K. and M.Y.I., Re-source, S.A.K., A.B.M.; Project administration, A.B.M.; Obtaining funding, A.B.M. All authors have read and agreed to the pub- lished version of the manuscript.
Line 777 – wrong citation! – author of cited paper is not Ruslan Kalendar only, but O. Khapilina and other 6 authors (including R. Kalendar).
Authors’ response: According to the reviewer's recommendations, appropriate changes have been made
Khapilina, O.; Turzhanova, A.; Danilova, A.; Tumenbayeva, A.; Shevtsov, V.; Kotukhov, Y.; Kalendar, R. Primer Binding Site (PBS) Profiling of Genetic Diversity of Natural Populations of Endemic Species Allium ledebourianum Schult. BioTech 2021, 10, 23.
https://doi.org/10.3390/biotech10040023
ENGLISH
English of the manuscript is often using inappropriate terms and repeatedly it is very difficult to follow. Examples of inappropriate terms are:
- line 73 – not spreading, but distribution
- line 297 – not “leading families”, but “major families” or “the commonest families”
Example of sentence, which requires considerable rewriting is on lines 315-316 (“amplification of P. anomala DNA resulted in clearly distinguishable amplicons, varied in number”) [Did authors mean “Amplified PCR products varied in number among different primers, or among samples, or what?]. Or lines 72-75 (“hearts of spreading” [Did authors mean “centre of distribution”?], “attribute to less extensive areas of distribution”). And unfortunately, throughout the whole manuscript.
Above mentioned examples are just few of many I strongly suggest to submit paper to some professional Language Editing service.
Authors’ response: We have completely revised the text and improved the Language Editing

Round 2
Reviewer 1 Report
Comments and Suggestions for Authors
The article has been revised extensively and this is definitely a better version than the previous one. I appreciate the tremendous work the authors have put into the current form of the manuscript.
My main complaint still relates to the use of iPBS molecular markers to estimate genetic variation. The authors are uncritical of this method in trying to convince me of it, and placing it above SSR.
I have a lot of experience with AFLP and RAPD, and there is a reason why these markers are virtually unused and unpublished anymore. RAPD is known to be a capricious marker (to say the least) and reagents have been discontinued for AFLP, even though analysis on a sequencer gave quite good multilocus results.
I am not persuaded that analysis based only on bands reads on agarose gel are better and more efficient than using automated sequencers.
I do not agree with such opinions of the authors as they give in their response to my review or directly in the text:
->The selection of PCR markers using PBS primers to identify genetic polymorphism is more effective than other existing PCR approaches, including simple sequence repeats (SSR) markers.
-> The method's reproducibility may be attributed to the standardized PCR reaction conditions in our country. If other authors adopt the exact protocol we have proposed, similar results should be achieved in their laboratories.
-> This method is universal because interspersed repeats and transposable elements are present in all eukaryotic species, making it easily adaptable to any plant species, including Paeonia sp. The study of genetic polymorphism in P. anomala using SSR markers is less appealing to us, as we aim to avoid duplicating the work of other researchers
In the text:
->Line 471-474 What does well-defined bands mean and what, by the way, to the trash?
To study the genetic variability of P. anomala wild populations, only clear well-defined bands were considered since each band of unique size corresponds to a unique locus. The presence or absence of a band in the amplification spectrum of a particular genotype was used to construct a binary matrix. Reproducible fragments were scored as present (1) or absent (0).
-> the amplification spectra/spectrum - inappropriate vocabulary
In summary, the discussion on genetic diversity is mainly about convincing us that the iPBS markers used are a great choice. I would look at them more critically and apply myself to writing this part of the discussion again.
I was not convinced by the authors.
Author Response
Comments and Suggestions for Authors
The article has been revised extensively and this is definitely a better version than the previous one. I appreciate the tremendous work the authors have put into the current form of the manuscript.
Author's response:
We thank to review for the work done and for the great help provided to the authors in the work on the manuscript.
My main complaint still relates to the use of iPBS molecular markers to estimate genetic variation. The authors are uncritical of this method in trying to convince me of it, and placing it above SSR. I have a lot of experience with AFLP and RAPD, and there is a reason why these markers are virtually unused and unpublished anymore. RAPD is known to be a capricious marker (to say the least) and reagents have been discontinued for AFLP, even though analysis on a sequencer gave quite good multilocus results. I am not persuaded that analysis based only on bands reads on agarose gel are better and more efficient than using automated sequencers.
Author's response:
The RAPD method technology is based only on the feature of Taq-polymerase, which can use a primer with binding errors to initiate elongation. But only under the condition that the primer concentration is high and the annealing temperature is low.
The iPBS method is also a variant of the RAPD method. But we also suggest using a much lower primer concentration in this protocol to increase the specificity of the method, and an annealing temperature of 55°C or higher. Therefore, non-reproducibility of amplification profiles will be kept to a minimum, although partially possible. The polymorphism revealed by this method is high and applicable for intraspecific analysis, it may be better than other approaches, due to the fact that this is a simple and accessible method, simpler than AFLP and there is no saturation of genetic polymorphism, as with AFLP.
I do not agree with such opinions of the authors as they give in their response to my review or directly in the text:
->The selection of PCR markers using PBS primers to identify genetic polymorphism is more effective than other existing PCR approaches, including simple sequence repeats (SSR) markers.
Author's response:
We think that the study of polymorphism using PBS markers is one of the alternative methods, as well as using SSR, ISSR, AFLP markers. The advantage of this method is its universality and the possibility of using it on any eukaryotic organisms. A number of works also show the direct involvement of some mobile elements in the adaptation of plants to stress. We think that using this type of markers it is possible to identify genetic changes that occur in the genome due to the impact of various stress factors in wild populations of P.anomala living in the same region with different environmental conditions.
-> The method's reproducibility may be attributed to the standardized PCR reaction conditions in our country. If other authors adopt the exact protocol we have proposed, similar results should be achieved in their laboratories.
Author's response:
All studies on the analysis of PBS polymorphism (amplification, DNA fingerprint) were carried out twice for each DNA sample, and this gives confidence in the reproducibility of this method. The use of standardized amplification protocols will allow other authors to obtain similar results. However, depending on PCR conditions, for example, the use of Taq-polymerase or higher concentrations of primers will lead to a completely different or similar spectrum of amplicons. But within a specific protocol, the results will be reproducible and effective.
-> This method is universal because interspersed repeats and transposable elements are present in all eukaryotic species, making it easily adaptable to any plant species, including Paeonia sp. The study of genetic polymorphism in P. anomala using SSR markers is less appealing to us, as we aim to avoid duplicating the work of other researchers
Author's response:
The SSR method is good only if a lot of loci are detected, and the frequencies of which are close. Such SSR loci are unique and rare. However, genomic analysis requires hundreds or even thousands of similar loci at hand to analyze a variety of material. But this is impossible for rare species, but only for cultivated plants of breeding or nutritional interest.
In the text:
->Line 471-474 What does well-defined bands mean and what, by the way, to the trash?
To study the genetic variability of P. anomala wild populations, only clear well-defined bands were considered since each band of unique size corresponds to a unique locus. The presence or absence of a band in the amplification spectrum of a particular genotype was used to construct a binary matrix. Reproducible fragments were scored as present (1) or absent (0).
Author's response:
We agree with the reviewer, incorrect expression «well-defined bands», changed to Only clear scorable bands were used for studying genetic variability
-> the amplification spectra/spectrum - inappropriate vocabulary
Author's response:
We agree with the reviewer, incorrect expression. Changed to «the amplification profiles»
In summary, the discussion on genetic diversity is mainly about convincing us that the iPBS markers used are a great choice. I would look at them more critically and apply myself to writing this part of the discussion again.
Author's response:
We took into account the reviewer’s opinion and made additions to the “Discussion” section.

Reviewer 3 Report
Comments and Suggestions for Authors
The revised manuscript is much improved: the structure of paper was corrected, language editing/corrections of typing errors and mistakes was done responsibly, major comments regarding methodology and presentation of data were accepted.
I do not have further comments on the paper. There are only few (very) minor issues remaining, which might be corrected in order to have the paper ready for publishing:
1) Figures 1 and 3 are highly informative as they are collage of images of same information. I strongly recommend to merge figures 4+5+6+7+8+9 in some similar kind of 1 collage or 2 collages. It is very similar information presented in these figures, it presents negative results and it covers too much space that disturbs reading/interrupts the text. Having all these images together will make data more synoptic.
2) Figure 2 is now much better. However, it still has a minor error in its legend. According to ISO-3166 country codes is the correct abbreviation for China "CN" (not CH, which is Switzerland; https://en.wikipedia.org/wiki/List_of_ISO_3166_country_codes)
3) Line 146-147: "data are available from the author upon REASONABLE request" sounds strange. I recommend to write just "... upon request" and let the author decide whether the request is reasonable or not, but let it be more "Open Science".
Author Response
The revised manuscript is much improved: the structure of paper was corrected, language editing/corrections of typing errors and mistakes was done responsibly, major comments regarding methodology and presentation of data were accepted.
Author's response:
We thank to review for the work done and for the great help provided to the authors in the work on the manuscript.
I do not have further comments on the paper. There are only few (very) minor issues remaining, which might be corrected in order to have the paper ready for publishing:
- Figures 1 and 3 are highly informative as they are collage of images of same information. I strongly recommend to merge figures 4+5+6+7+8+9 in some similar kind of 1 collage or 2 collages. It is very similar information presented in these figures, it presents negative results and it covers too much space that disturbs reading/interrupts the text. Having all these images together will make data more synoptic.
Author's response:
The authors agree with the remark, the drawings were combined into collages.
- Figure 2 is now much better. However, it still has a minor error in its legend. According to ISO-3166 country codes is the correct abbreviation for China "CN" (not CH, which is Switzerland; https://en.wikipedia.org/wiki/List_of_ISO_3166_country_codes)
Author's response:
Thank you! We've corrected CH to CN
3) Line 146-147: "data are available from the author upon REASONABLE request" sounds strange. I recommend to write just "... upon request" and let the author decide whether the request is reasonable or not, but let it be more "Open Science".
Author's response:
We have corrected this sentence as recommended by the reviewer.
